# Verifying Chain-of-Thought Reasoning via Its Computational Graph

**Zheng Zhao**[1,3][*] **Yeskendir Koishekenov**[1] **Xianjun Yang**[2] **Naila Murray**[1] **Nicola Cancedda**[1]
[1]FAIR at Meta  [2]Meta Superintelligence Labs  [3]University of Edinburgh
`zheng.zhao@ed.ac.uk, yeskendir@meta.com, xianjunyang@ucsb.edu`
`murrayn@meta.com, canceddan@acm.org`

## Abstract

Current Chain-of-Thought (CoT) verification methods predict reasoning correctness based on outputs (black-box) or activations (gray-box), but offer limited insight into *why* a computation fails. We introduce a white-box method: **Circuit-based Reasoning Verification (CRV)**. We hypothesize that attribution graphs of correct CoT steps, viewed as *execution traces* of the model's latent reasoning circuits, possess distinct structural fingerprints from those of incorrect steps. By training a classifier on structural features of these graphs, we show that these traces contain a powerful signal of reasoning errors. Our white-box approach yields novel scientific insights unattainable by other methods. (1) We demonstrate that structural signatures of error are highly predictive, establishing the viability of verifying reasoning directly via its computational graph. (2) We find these signatures to be highly domain-specific, revealing that failures in different reasoning tasks manifest as distinct computational patterns. (3) We provide evidence that these signatures are not merely correlational; by using our analysis to guide targeted interventions on individual transcoder features, we successfully correct the model's faulty reasoning. Our work shows that, by scrutinizing a model's computational process, we can move from simple error detection to a deeper, causal understanding of LLM reasoning.

## 1 Introduction

Chain-of-Thought (CoT; Wei et al., 2022; Kojima et al., 2022) prompting has proven to be a powerful method for boosting the performance of Large Language Models (LLMs). This capability is now central to the latest generation of reasoning models, such as DeepSeek-R1 (DeepSeek-AI et al., 2025) and OpenAI's o1 (OpenAI et al., 2024). Despite this success, a fundamental vulnerability persists across the spectrum of these systems: the reasoning process itself is sometimes flawed (Turpin et al., 2023; Li et al., 2025b; Arcuschin et al., 2025; Lindsey et al., 2025; Chen et al., 2025b).

This reliability gap has spurred research into automated verification. Current methods fall into two main categories. *Black-box* approaches analyze the generated text or final logit distribution (Jacovi et al., 2024; Wang et al., 2025b; Baker et al., 2025). *Gray-box* approaches look at the model's internal state, using simple probes on raw activations or analyzing the trajectory of hidden states (Xie et al., 2025; Zhang et al., 2025; Afzal et al., 2025; Bi et al., 2025; Wang et al., 2025a). While insightful, these methods are fundamentally limited; they can *detect* that a model's internal state is correlated with an error, but not *explain why* the underlying computation leads to an error.

This limitation motivates a deeper, more mechanistic approach. We postulate that models implement latent algorithms that solve specific tasks through specialized subgraphs, or **circuits** (Olah et al., 2020; Elhage et al., 2021). From this perspective, a reasoning failure is not merely an erroneous state, but a flaw in the execution of a latent algorithm. To diagnose such flaws requires inspecting the underlying computational process, akin to examining an *execution trace* in classical software. We propose to approximate this trace by constructing an **attribution graph** (Dunefsky et al., 2025)—a structural representation of the causal information flow between model components.

---

[*]Work done during an internship at Meta FAIR.

For such a graph to serve as a meaningful trace, its components must be interpretable. We therefore first create an interpretable surrogate model by replacing its standard MLP modules with trained **transcoders** (Dunefsky et al., 2025). We then construct and analyze attribution graphs over the sparsely activating features of such surrogate model (Ameisen et al., 2025). Finally, to formally test whether these traces contain a detectable signal of error, we train a diagnostic classifier on their structural properties. This entire methodology, which we call **Circuit-based Reasoning Verification (CRV)**, is thus designed as a scientific instrument to investigate our central hypothesis: *that reasoning failures manifest as detectable structural signatures on their computational execution traces, which can be leveraged for automated verification.*

As a scientific instrument, CRV requires a controlled experimental setting. While advanced reasoning models employ complex mechanisms like search and backtracking, their convoluted reasoning paths can obscure the fundamental computations of a single reasoning step. Our work therefore focuses on standard, instruction-tuned models generating autoregressive CoT, as this paradigm provides a clearer window into the primitive computations that underpin emergent reasoning. While our approach, despite being effective, is too computationally intensive to be intended as a practical, drop-in verifier, it yields novel scientific insights unattainable by other methods. Our main contributions are therefore not just about performance, but about understanding:

- We introduce Circuit-based Reasoning Verification, a white-box method for analyzing reasoning failures, showing that verifying reasoning via its computational graph is feasible.

- We find that the structural signatures of error are highly domain-specific, revealing that failures in executing different reasoning tasks manifest as distinct computational patterns.

- We establish the causal role of these error signatures, successfully correcting faulty reasoning via targeted interventions on individual transcoder features.

- To support future research, we release datasets with step-level correctness labels for CoT reasoning on synthetic and real-world tasks, along with our trained transcoders.[1]

## 2 PROBLEM FORMULATION AND PRELIMINARIES

### 2.1 PROBLEM STATEMENT

Let an LLM generate a Chain-of-Thought $S = (s_1, s_2, \ldots, s_m)$ to solve a problem, where each step $s_i$ is a sequence of tokens. During the generation of step $s_i$, the underlying model produces a computational state $\mathcal{M}_i$. From this state, we construct an **attribution graph** $G_i = (\mathcal{V}, \mathcal{E})$, where vertices $\mathcal{V}$ represent interpretable features and tokens, and edges $\mathcal{E}$ represent the causal influence between them (see Section 3.2). From each graph $G_i$, we extract a fixed-size feature vector $\mathbf{x}_i = \phi(G_i)$, where $\phi$ is a feature extraction function designed to capture the graph's structural properties. We term this vector the step's *structural fingerprint*. Our goal is to learn a diagnostic classifier $f_\theta$ that takes this structural fingerprint as input to predict the correctness of the reasoning step:

$$\hat{y}_i = f_\theta(\mathbf{x}_i)$$

where $\hat{y}_i \in \{\text{correct}, \text{incorrect}\}$.

### 2.2 PRELIMINARIES: CIRCUITS IN TRANSFORMERS

The term "circuit" in mechanistic interpretability refers to a specific subgraph within a neural network that implements a human-understandable algorithm (Olah et al., 2020). In Transformers (Vaswani et al., 2017), these circuits are composed of attention heads and MLP computations. Our work is conceptually motivated by the prospect of finding patterns distinguishing sound and faulty activations of circuits involved in reasoning. While our method does not observe these circuits directly, our hypothesis is that they cast detectable *structural fingerprints* onto the attribution graphs we construct. A primary goal of our subsequent analysis is therefore to interpret the graph-based features that are most predictive of failure as the signatures of these underlying error patterns.

---

[1]`https://github.com/facebookresearch/CRV`

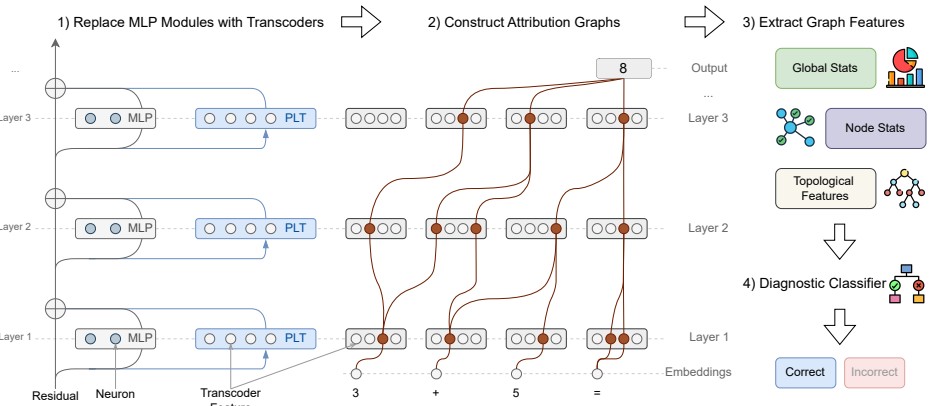

Figure 1: The CRV pipeline. (1) The LLM's MLP modules are replaced with per-layer transcoders (PLTs), making it interpretable. (2) For a given CoT step, we generate an attribution graph capturing causal flow between interpretable features and model components. (3) Structural features are extracted from this graph, and (4) fed to a diagnostic classifier to predict the step's correctness.

## 2.3 PRELIMINARIES: TRANSCODERS FOR INTERPRETABLE FEATURES

A significant challenge in analyzing model activations is their high dimensionality and lack of direct interpretability. A powerful approach to this challenge is to learn a sparse, overcomplete basis for these activations using a sparse autoencoder (SAE; Cunningham et al., 2023). An SAE is trained to reconstruct an activation vector $x \in \mathbb{R}^d$ from a much higher-dimensional, but mostly zero, feature vector $f \in \mathbb{R}^D$, where $D \gg d$. The elements of $f$ correspond to a set of learned, interpretable features, sparsely activated by inputs. While the canonical SAE objective is to reconstruct its own input ($f(x) \approx x$), our work leverages a variant known as a transcoder (Dunefsky et al., 2025), which is instead trained to approximate the *input-output function* of a target component, such as an MLP ($f(x) \approx \text{MLP}(x)$). This approach makes the transcoder a true *functional substitute* for the original module. Its objective is not mere reconstruction, but the emulation of a computational step in an interpretable, sparsely activated basis. By replacing a model's standard MLP module with a trained transcoder, we force its intermediate computations to be represented not by a dense vector, but by a sparse combination of these meaningful features.

## 3 METHODOLOGY

Unlike in Process Reward Modeling (PRM), where the goal is limited to judging the correctness of a reasoning step, we take the perspective of a model developer interested in debugging reasoning failures in a specific model to which they have full access. We introduce **Circuit-based Reasoning Verification (CRV)**, a method for detecting flawed reasoning by analyzing its structural fingerprint.

### 3.1 DATASET CURATION AND STEP-LEVEL ANNOTATION

A prerequisite for developing our method is a dataset with reliable step-level correctness labels. Furthermore, our white-box methodology imposes a critical requirement that distinguishes our data needs from prior work. Since CRV analyzes the causal computational graph that produces a reasoning step, we must capture the full internal state of our specific model during the generation process. Consequently, existing text-only datasets such as PRM800K (Lightman et al., 2024) and REVEAL (Jacovi et al., 2024), which provide static '(text, label)' pairs and are designed for training black-box verifiers, are incompatible with our mechanistic approach. We must generate and label our own model's CoT outputs to create the necessary '(text, label, computational trace)' tuples for analysis. We therefore created a new benchmark covering both controlled synthetic tasks and the real-world GSM8K dataset (Cobbe et al., 2021).

**Synthetic Datasets (Boolean and Arithmetic).** To study reasoning failures in a controlled environment, we generated two datasets. The first involves evaluating complex boolean expressions, while the second involves multi-step arithmetic problems. The motivation for these datasets is the unam-

biguous ground truth: the correctness of any step in the reasoning chain (e.g., "15 + 7 = 22") can be verified automatically by a simple parser and evaluator. This allows us to generate a large, labeled dataset for initial training and analysis. Furthermore, these tasks are intrinsically compositional, and the complexity of samples can be fully controlled. Further details are provided in Appendix A.

**Step-Level Annotation for GSM8K.** Annotating a real-world dataset like GSM8K is challenging. To scale, we used a semi-automated process with a stronger LLM (e.g., Llama 3.3 70B Instruct) as an expert judge. For each CoT, the judge evaluated step correctness given the full problem context. We validated these labels through manual review of a substantial subset, yielding a high-fidelity dataset for real-world reasoning. Further details are provided in Appendix A.

## 3.2 Circuit-based Reasoning Verification (CRV)

CRV is a four-stage pipeline designed to classify the correctness of a CoT step by analyzing the computational graph of a modified, interpretable LLM. An overview is presented in Figure 1.

### 3.2.1 Step 1: Replacing MLPs with Interpretable Transcoders

The foundation of CRV is an architectural modification that makes the target LLM interpretable. For each MLP module in the model, we train a corresponding transcoder on a large, diverse dataset of activations harvested from the original LLM.[2] The training objective combines an L2 reconstruction loss with a TopK activation function, which enforces sparsity by preserving only the k-largest feature activations. Once trained, we replace the MLP module for each layer in the LLM with its corresponding transcoder. The forward pass of the model is now forced to flow through these sparse, interpretable bottlenecks. All subsequent analysis is performed on this modified, interpretable replacement model. Full details of the transcoder architecture and training are provided in Appendix B.

### 3.2.2 Step 2: Constructing Step-Level Attribution Graphs

With our transcoder-infused replacement model, we require a principled method to trace information flow and construct a causal graph of the computation. To this end, we adapt the recent circuit analysis methodology of Dunefsky et al. (2025). Applying their greedy path-finding algorithm allows us to trace high-attribution connections backward from the final logits, yielding a sparse, weighted, directed graph $G_i = (\mathcal{V}, \mathcal{E})$ for each reasoning step $s_i$. This graph represents the core computational subgraph, where the nodes $\mathcal{V}$ are the disjoint union of input tokens, active transcoder features, and output logits. The directed edges $\mathcal{E}$ represent the high-attribution causal pathways between these components (e.g., from an early-layer feature to a later-layer feature, or from a feature to a logit), with weights quantifying the strength of their influence. For a complete derivation and description of the circuit-finding algorithm, we refer the reader to the original work (Dunefsky et al., 2025).[3]

### 3.2.3 Step 3: Extracting Interpretable Graph Features

From each attribution graph $G_i$, we extract a fixed-size feature vector $\mathbf{x}_i$ as a structural fingerprint of the computation. We prune the graph to its most influential components, retaining nodes and edges accounting for a threshold (e.g., 80%) of total influence to the final logits. The feature set, calculated on this pruned subgraph (unless stated otherwise), is organized into three hierarchical levels.

**Global Graph Statistics:** These features capture a high-level summary of the computational subgraph, including the count of active feature nodes after pruning and the final logit probability and entropy. They provide a coarse measure of the computation's complexity and uncertainty.

**Node Influence and Activation Statistics:** This group quantifies the properties of the interpretable feature nodes. We compute statistics (mean, max, std) on their activation values and influence scores. This helps distinguish computations driven by a few highly active, decisive features from those driven by a diffuse combination of many weak features. We also include a histogram of active features by layer, which characterizes the computational depth of the reasoning step.

---

[2]This is also referred as per-layer transcoders (PLTs) by Ameisen et al. (2025).

[3]We use implementation from Hanna et al. (2025) for computing attribution graphs in our work.

**Topological and Path-Based Features:**    To analyze the structure of the information flow, we compute a rich set of topological features on the pruned subgraph. These include graph density, centrality measures (degree, betweenness) to identify computational hubs, and connectivity metrics.

This comprehensive feature set provides the foundation for our diagnostic classifier. A full list and detailed motivation for each feature is provided in Appendix C.1.

### 3.2.4    STEP 4: DIAGNOSTIC CLASSIFIER

For the final classification step, we use a Gradient Boosting Classifier (GBC) trained on the extracted feature vectors: $f_\theta(\mathbf{x}_i) = \hat{y}_i$. GBC suits for our heterogeneous, tabular features and provides robust feature importance measures, which we leverage to identify the most predictive structural properties of error circuits. We also benchmark against several alternative classifiers in Appendix C.4.

## 4    EXPERIMENTS

We conduct a series of experiments designed to validate the central hypothesis of our work: *that the attribution graphs of reasoning steps contain a rich, structural signal of their correctness*. Our evaluation is structured around three primary research questions. First, we investigate whether CRV's white-box approach significantly outperforms a comprehensive suite of gray-box and black-box baselines in verification accuracy and test its robustness to domain shifts and increasing task difficulty **(RQ1)**. Next, we analyze our trained models to identify which specific computational structures within the graph are most predictive of failure, moving from detection to mechanistic understanding **(RQ2)**. Finally, we conduct exploratory studies to assess if these mechanistic insights can be used to perform targeted, causal interventions that correct faulty reasoning **(RQ3)**.

### 4.1    EXPERIMENTAL SETUP

**Models and Datasets.**    Our experiments are conducted on the **Llama 3.1 8B Instruct** model (AI@Meta, 2024). We select the instruction-tuned variant, as its prompt-following optimization is critical for reliably eliciting the CoT reasoning traces for our analysis. This model is then modified with our trained transcoders as described in Section 3. We evaluate performance on our three datasets: Synthetic (Boolean), Synthetic (Arithmetic), and the annotated GSM8K benchmark.

**Baselines.**    We compare CRV against two categories of baselines. First, black-box methods that use the final logit distribution: Maximum Softmax Probability (MaxProb), Perplexity (PPL), Entropy, Temperature Scaling (Temp. Scaling; Shih et al., 2023), and Energy (Liu et al., 2020). Second, gray-box methods that operate on internal states. This includes trajectory-based methods that analyze hidden state dynamics across layers, such as Chain-of-Embedding (with its real-space CoE-R and complex-space CoE-C variants; Wang et al., 2025a) and CoT-Kinetics (Bi et al., 2025), as well as a standard logistic regression probe (LR Probe) and a two-layer multilayer perceptron probe (MLP Probe) trained on the step's average hidden state.[4]  While CoE and CoT-Kinetics were originally designed for full CoT evaluation, they prove to be strong step-level baselines. All implementation details are deferred to Appendix C.2.

**Evaluation Metrics.**    We use AUROC, FPR@95, and AUPR to evaluate verifier performance. As our goal is the detection of reasoning failures, we treat the incorrect label as the positive class for all metric calculations. AUROC assesses how well the method ranks correct versus incorrect steps across thresholds. AUPR captures the precision-recall trade-off for the positive (incorrect) class. FPR@95 measures the false positive rate when 95% of positives are correctly identified, reflecting reliability under strict conditions; a lower score indicates the verifier can detect most errors with minimal false alarm. Together, these metrics provide complementary views of performance. Precise mathematical formulations and implementation details for all metrics are provided in Appendix C.3.

---

[4]We also evaluated a last-token probe, but found that using the average representation yielded slightly better performance.

Table 1: Verification performance. Arrows indicate preferred direction (↑ higher is better, ↓ lower is better). **Best** and second-best results are highlighted for each metric. The low AUPR on the Boolean dataset reflects extreme label imbalance, with the incorrect label only 0.2% (Appendix A.5).

| Paradigm | Method | Synthetic (Boolean) | | | Synthetic (Arithmetic) | | | GSM8K | | |
|---|---|---|---|---|---|---|---|---|---|---|
| | | AUROC ↑ | AUPR ↑ | FPR@95 ↓ | AUROC ↑ | AUPR ↑ | FPR@95 ↓ | AUROC ↑ | AUPR ↑ | FPR@95 ↓ |
| *Black-Box* | MaxProb | 58.81 | 0.34 | 95.20 | 61.87 | 1.81 | 84.98 | 54.91 | 7.99 | 91.86 |
| | PPL | 57.37 | 0.29 | 91.02 | 60.19 | 1.68 | 85.52 | 55.46 | 8.12 | 90.69 |
| | Entropy | 53.56 | 0.24 | 97.55 | 60.03 | 1.52 | 85.40 | 56.67 | 7.29 | 87.08 |
| | Temp. Scaling | 58.77 | 0.36 | 91.41 | 59.67 | 1.66 | 86.96 | 54.42 | 8.24 | 92.28 |
| | Energy | 51.08 | 0.28 | 95.11 | 76.45 | 5.59 | 73.86 | 62.55 | 9.11 | 86.34 |
| *Gray-Box* | CoE-R | 53.17 | 0.33 | 92.85 | 58.47 | 1.93 | 76.68 | 52.38 | 8.34 | 96.20 |
| | CoE-C | 51.03 | 0.38 | 92.07 | 69.39 | 3.03 | 63.33 | 53.57 | 10.80 | 96.33 |
| | CoT-Kinetics | 53.62 | 0.24 | 97.13 | 60.83 | 1.58 | 85.09 | 56.54 | 7.35 | 86.83 |
| | LR Probe | 52.91 | 0.25 | 88.42 | 54.22 | 1.50 | 91.90 | 55.86 | 7.99 | 90.32 |
| | MLP Probe | 53.63 | 0.26 | 88.56 | 54.41 | 1.30 | 90.98 | 56.02 | 8.63 | 93.94 |
| **White-Box** | **CRV (Ours)** | **75.87** | **0.97** | **79.17** | **92.47** | **28.92** | **37.09** | **70.17** | **14.3** | **79.61** |

## 4.2 Verification Performance and Robustness (RQ1)

We first address RQ1 by evaluating CRV against all baselines on the task of reasoning step verification and then probing its robustness under more challenging conditions.

**Main Verification Performance.** The results, presented in Table 1, provide strong empirical support for our central hypothesis: that the structural signatures present in a reasoning step's computational trace contain a directly verifiable signal of its correctness. CRV consistently outperforms all black-box and gray-box baselines across every dataset and metric. The strength of this structural signal is particularly evident on the synthetic datasets. On the Arithmetic task, for instance, CRV achieves an AUROC of 92.47, a significant leap over the strongest baseline score of 76.45. This advantage in reliability is further underscored by the FPR@95, where CRV reduces the false positive rate to 37.09% from the baseline's 63.33%. The performance gap is most pronounced on these structured, synthetic datasets. We hypothesize that the structured nature of algorithmic reasoning induces highly consistent execution traces for valid solutions. Consequently, the structural signatures of error manifest as more uniform deviations from this baseline, rendering them highly detectable.

**Analysis of Cross-Domain Generalization.** A key difference between CRV and most baselines is that its diagnostic classifier requires training. A critical question, therefore, is whether CRV learns domain-specific correlations or more fundamental, generalizable signatures of flawed reasoning. To test this, we conduct a comprehensive cross-domain evaluation. We train a CRV classifier on each of our three datasets individually and evaluate its zero-shot performance on the other two unseen tasks. We also evaluate a model trained on the union of all training sets to test if a single classifier can learn diverse error patterns simultaneously.

Table 2 shows that CRV's zero-shot transfer is limited, suggesting that error fingerprints are highly task-specific. For example, CRV trained solely on Arithmetic achieves an AUROC of 57.04 on GSM8K, falling short of the Energy baseline. This reveals that errors in different reasoning tasks (e.g., boolean logic vs. natural language arithmetic) produce distinct structural patterns in the computational graph.

Table 2: Cross-domain generalization performance. For each test dataset, we compare the strongest baseline (based on AUROC) against CRV trained *in-domain* and out-of-domain. **Best** out-of-domain results are highlighted.

| Test Set | Method (Train Set) | Metrics | | |
|---|---|---|---|---|
| | | AUROC ↑ | AUPR ↑ | FPR@95 ↓ |
| Boolean | Baseline (MaxProb) | 58.81 | 0.34 | 95.20 |
| | CRV (GSM8K) | 45.77 | 0.21 | 97.28 |
| | CRV (Arithmetic) | **61.58** | **0.51** | **87.55** |
| | CRV (Combined) | 65.18 | 0.65 | 80.15 |
| | *CRV (Boolean)* | *75.87* | *0.97* | *79.17* |
| Arithmetic | Baseline (Energy) | **76.45** | **5.59** | 73.86 |
| | CRV (GSM8K) | 55.11 | 1.50 | 91.91 |
| | CRV (Boolean) | 69.59 | 2.64 | **72.87** |
| | CRV (Combined) | 90.51 | 30.57 | 44.23 |
| | *CRV (Arithmetic)* | *92.47* | *28.92* | *37.09* |
| GSM8K | Baseline (Energy) | **62.55** | **9.11** | **86.34** |
| | CRV (Boolean) | 44.37 | 6.33 | 95.71 |
| | CRV (Arithmetic) | 57.04 | 7.85 | 94.37 |
| | CRV (Combined) | 70.62 | 12.66 | 80.22 |
| | *CRV (GSM8K)* | *70.17* | *14.3* | *79.61* |

However, this domain specificity does not preclude the creation of a robust, multi-task verifier. As shown in the CRV (Combined) rows, when the classifier is exposed to diverse error signatures during

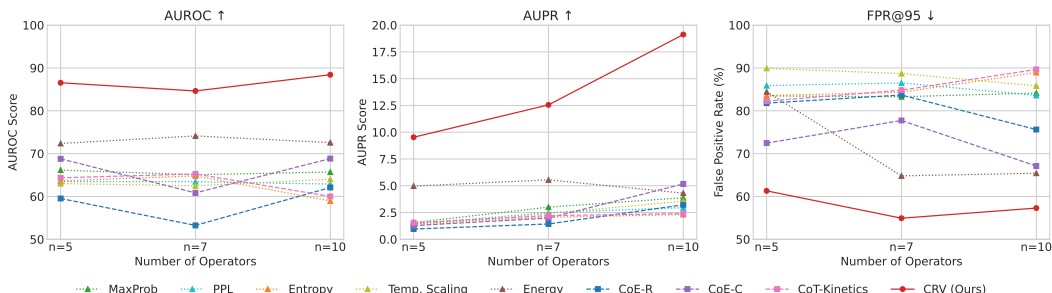

Figure 2: Performance of the step correctness predictors on the synthetic arithmetic task as a function of difficulty (number of operators). CRV retains a clear advantage as complexity increases.

training, performance rebounds. On the Arithmetic task, the combined model achieves an AUROC of 90.51, nearly matching the specialist in-domain model (92.47) and outperforming baselines. Remarkably, on GSM8K, the combined model (70.62 AUROC) slightly outperforms the in-domain specialist (70.17). This suggests that while the structural boundaries of error are distinct across tasks, they are not incompatible; a single classifier can learn the union of these complex "failure geometries", and in the case of GSM8K, potentially benefit from the inductive bias provided by synthetic logical tasks.

**Performance Under Increasing Difficulty.**   To further probe CRV's robustness, we analyze its performance on the synthetic arithmetic dataset as a function of problem complexity, controlled by the number of operators ($n \in \{5, 7, 10\}$).[5]   Figure 2 plots the performance of CRV against key baselines across these difficulty levels. While most methods show stable AUROC and FPR@95, CRV maintains a consistent advantage across all difficulty levels. AUPR generally improves for all methods as difficulty rises because harder problems increase the proportion of incorrect examples (a condition to which AUPR is sensitive). Importantly, CRV's advantage persists despite these shifts, highlighting the robustness of its structural signals across task difficulty and class balance.

### 4.3 MECHANISTIC ANALYSIS OF ERROR COMPUTATIONS (RQ2)

Having demonstrated CRV's predictive power, we now turn to its key advantage: interpretability. To address RQ2, we dissect our graph representation to identify structural "fingerprints" of error, from high-level feature ablation to fine-grained analysis of the most predictive structures.

**Ablation of Feature Families.**   A leave-one-out ablation study on the Synthetic (Arithmetic) dataset reveals a clear hierarchy of feature importance, as summarized in Table 3. The *Node Influence & Activation* features are demonstrably the most critical; their removal causes the most performance degradation across all metrics, most notably increasing FPR@95 by over 12 points. The *Global Graph Statistics* also provide a substantial contribution. Interestingly, the *Topological & Path-Based* features

Table 3: Leave-one-out ablation study on the Synthetic (Arithmetic) dataset.

| Feature Set | Arithmetic | | |
|---|---|---|---|
| | AUROC ↑ | AUPR ↑ | FPR@95 ↓ |
| **CRV (All three families)** | **92.47** | **28.92** | **37.09** |
| *Ablation:* | | | |
| – w/o Global Stats | 89.62 | 24.35 | 44.54 |
| – w/o Node Stats | 88.31 | 23.25 | 49.07 |
| – w/o Topological Stats | 90.89 | 26.83 | 39.19 |

appear least critical for this specific task, suggesting that the state of key local features is a more dominant signal than the holistic graph structure. Nevertheless, the full CRV model, which integrates all three signal types, is required to achieve optimal verification performance.

**Visualizing the Structural Signatures of Error.**   To provide qualitative evidence for our hypothesis, we visualize the "structural fingerprints" learned by our classifier. Figure 4 shows distributions of five highly predictive features for correct versus incorrect GSM8K reasoning steps. Across diverse feature types, from graph topology (e.g., *Graph Density*) to node statistics (e.g., *Total Active*

---

[5]We exclude $n = 3$ as the model's high accuracy yields too few incorrect examples for reliable evaluation.

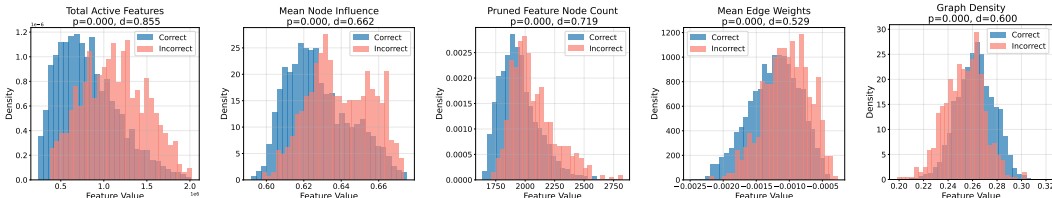

Figure 4: **Topological Fingerprints of Error on GSM8K.** Distributions of five selected graph features for correct (blue) vs. incorrect (red) reasoning steps. The visual separation is statistically significant for each feature shown (independent t-test, $p < 0.001$) and represents a medium-to-large effect size (Cohen's d). This provides quantitative evidence that attribution graphs contain a clear, separable structural signal of a computation's integrity.

*Features*), distributions are clearly distinct. Similar patterns are observed on our synthetic datasets (see included in Appendix C.5), confirming that the graph representation captures separable structural differences between valid and flawed computations.

While individual features are predictive, CRV's strength lies in their combination. To illustrate this, we project the full high-dimensional feature vectors into two dimensions via Principal Component Analysis (PCA). Figure 3 reveals that incorrect steps form a dense subset *within* the broader distribution of correct steps. Crucially, correct steps also occupy a dis-

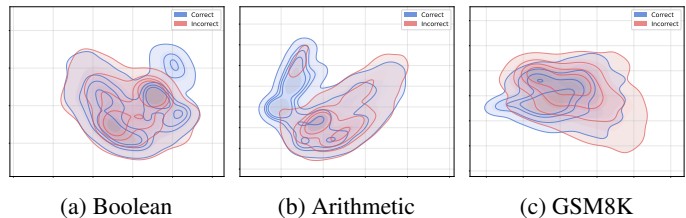

(a) Boolean      (b) Arithmetic      (c) GSM8K

Figure 3: Distributions of features after PCA for correct (blue) vs. incorrect (red) reasoning steps.

tinct region not shared by incorrect computations. This suggests many reasoning failures are computational "near misses", structurally similar to valid steps, which explains the overlap. Yet there exists a *zone of computational integrity*, a region defined by structural properties accessible only to correct reasoning. A complementary t-SNE visualization in Appendix C.5 shows the same structure, confirming that CRV succeeds by learning the complex boundary of this zone rather than separating two simple clusters.

### 4.4 FROM DIAGNOSIS TO CAUSAL INTERVENTION (RQ3)

Finally, we conduct an exploratory study to test whether CRV's diagnostic insights can guide prescriptive interventions. A key advantage of our white-box approach is that failure-predictive graph features can be traced to specific components of the model's computation. We illustrate this with a case study on the arithmetic task, correcting an order-of-operations error.

Given the expression `(7*((5+9)+7))`, the model incorrectly computed `7*14=98` in its second step (Table 4). CRV flagged this step as incorrect, and feature importance highlighted an unusually high activation of a late-layer transcoder feature. Tracing the signal back to the graph, we found a single highly active last-layer transcoder feature (ID 91814) strongly associated with multiplication.

We hypothesized that this premature activation caused the error, and performed a direct intervention. We re-ran the generation up to the point of failure and used a forward hook to manually clamp the activation of this specific *multiplication* feature to zero. The effect was immediate: as shown in the right column of Table 4, with the premature multiply impulse suppressed, the model correctly generated the next step `14+7=21` and proceeded to the correct final answer.

While not a general-purpose solution, this proof-of-concept is a crucial step. Beyond suppressing faulty features, we also corrected an error by *amplifying* an under-active feature (Appendix C.5). The success of both interventions provides closed-loop evidence that CRV's structural signatures are causally implicated in errors, opening a promising direction for targeted model interventions. Crucially, because these interventions rely on the semantic interpretation of specific transcoder fea-

Table 4: Side-by-side comparison of a reasoning trace before and after causal intervention. The highlight indicates the point of divergence where suppressing a single *multiplication* transcoder feature corrects the model's computational path.

| Before Intervention (Incorrect) | After Intervention (Correct) |
|---|---|
| Evaluate the arithmetic expression below. ( 7 * ( ( 5 + 9 ) + 7 ) ) To evaluate this expression, we need to follow the order of operations (PEMDAS): 1. Evaluate the expression inside the innermost parentheses: 5 + 9 = 14 2. Multiply 7 by the result: 7 * 14 = 98 3. Add 7 to the result: 98 + 7 = 105 Therefore, the expression evaluates to 105. | Evaluate the arithmetic expression below. ( 7 * ( ( 5 + 9 ) + 7 ) ) To evaluate this expression, we need to follow the order of operations (PEMDAS): 1. Evaluate the expression inside the innermost parentheses: 5 + 9 = 14 2. Add 7 to the result: 14 + 7 = 21 3. Multiply 7 by the result 7 * 21 = 147 Therefore, the value of the expression is 147. |

tures (e.g., identifying a missing "multiplication" concept), they demonstrate the potential of CRV to serve as a foundation for interpretable debugging, moving beyond simple error flagging to actionable model repair.

## 5 RELATED WORK

**CoT Prompting for Improved Reasoning.** Chain-of-Thought (CoT) prompting was introduced by Wei et al. (2022) to elicit more complex reasoning from LLMs through few-shot examples that demonstrate step-by-step problem solving. Subsequent work has expanded this idea in several directions. Kojima et al. (2022) showed that even a simple zero-shot instruction such as "Let's think step by step" can trigger coherent reasoning traces. While this reduces the need for handcrafted prompts, providing structured examples often remains beneficial. To scale this process, recent studies generate CoT exemplars synthetically (Zhang et al., 2023; Shao et al., 2023; Li et al., 2025a). Other work leverages test-time compute scaling to extend reasoning chains, enabling longer and more elaborate solutions (Snell et al., 2024). For comprehensive surveys of CoT techniques and their applications, see Chu et al. (2024) and Chen et al. (2025a).

**Verifying and Improving CoT Reasoning.** The transparency of CoT has also made it a focal point for research into model interpretability and reliability. While some work assumes reasoning traces are to some extent faithful representations of the model's internal process (Wei Jie et al., 2024; Korbak et al., 2025), a significant body of evidence highlights their unreliability (Arcuschin et al., 2025; Bentham et al., 2024; Chen et al., 2025b; Turpin et al., 2023). This has spurred a rich field of research dedicated to verifying and improving CoT traces. This research broadly investigates (i) the model's intrinsic ability to self-evaluate its reasoning steps (Zhang et al., 2025), (ii) how to measure the faithfulness of a reasoning chain to the final answer (Lanham et al., 2023; Bi et al., 2025; Tutek et al., 2025), and (iii) when reasoning steps are needed or useful (Bogdan et al., 2025; Wang et al., 2025b). A parallel line of work aims to improve reasoning chains through various forms of neuro-symbolic reasoning (Lyu et al., 2023), correction (Tyen et al., 2024), uncertainty calibration (Ji et al., 2025), or by enforcing internal consistency (Xie et al., 2025; Wang et al., 2025a). A distinct approach involves training auxiliary models, such as Process Reward Models (PRMs), to assess step-level correctness and guide post-training (Lightman et al., 2024; Wang et al., 2024; Guan et al., 2025). While all these methods aim to improve reasoning outcomes, they primarily operate on the textual or hidden state representations. We are not aware of previous attempts to verify reasoning by analyzing the structural properties of its underlying computational graph.

**Mechanistic Interpretability of CoT Reasoning.** Our work is most directly situated within the field of mechanistic interpretability, which seeks to reverse-engineer the algorithms learned by neural networks, moving beyond the surface-level analysis of CoT traces (Wei Jie et al., 2024; Korbak et al., 2025; Baker et al., 2025). A central tenet of this field is that models develop specialized subgraphs, or circuits, to perform specific computations (Olah et al., 2020). Recent work has begun to apply this lens to reasoning, providing evidence that CoT traces are driven by latent learned algo-

rithms (Venhoff et al., 2025) and even characterizing specific self-verification circuits in toy models (Lee et al., 2025). Other research aims to improve performance by eliciting or steering behavioral circuits (Zhao et al., 2025; Ward et al., 2025). A particularly powerful and increasingly popular tool in this area is the use of sparse autoencoders (SAEs), which learn to decompose a model's dense activation vectors into a sparse basis of interpretable features (Bricken et al., 2023; Cunningham et al., 2023). Our work builds directly on a variant, the transcoder (Dunefsky et al., 2025), which acts as a functional, interpretable substitute for an MLP module. While prior work has used transcoder-based attribution graphs to qualitatively analyze the faithfulness of CoT reasoning (Ameisen et al., 2025), our work is the first to operationalize this approach for automated verification. We move beyond visual inspection by systematically extracting quantitative, structural features from these graphs and demonstrating that they can be used to diagnose computational failures.

## 6 CONCLUSION

In this work, we introduced CRV, a white-box methodology for studying the computational structure of reasoning failures. By treating attribution graphs as execution traces of latent circuits, we showed that correct and incorrect reasoning leave distinct structural fingerprints. CRV revealed that these error signatures not only enable accurate verification but are also domain-specific, with failures in different reasoning tasks manifesting as distinct patterns. Moreover, targeted interventions on transcoder features demonstrated that these signatures are causally implicated, allowing us to correct faulty reasoning. Together, these findings establish CRV as a proof-of-concept for mechanistic analysis, showing that shifting from opaque activations to interpretable computational structure enables a causal understanding of how and why LLMs fail to reason correctly.

## ETHICS STATEMENT

Our research yields insights into success and failure patterns in LLM reasoning. Such knowledge could theoretically be used for malicious purposes, such as designing adversarial attacks or engineering more subtle, undetectable reasoning failures. However, the computationally intensive nature of CRV, which also requires deep expertise and white-box model access, positions it as a tool for deep scientific analysis rather than a scalable method for generating exploits. The primary and intended application of our work is defensive: by providing a scientific instrument for developers to diagnose why a model fails, we aim to accelerate the development of more robust, reliable, and safer AI systems. We believe the benefits of enabling a deeper, causal understanding of AI failures for safety and alignment research significantly outweigh the risks of misuse.

## REPRODUCIBILITY STATEMENT

We are committed to the reproducibility of our work. Our newly generated datasets with step-level labels and our trained transcoders are publicly available. We provide details on our experimental setup in Section 4.1. Comprehensive details are provided throughout the Appendix, including: our dataset construction, annotation prompts, and full data statistics (Appendices A.1, A.2, and A.5); our transcoder training procedure (Appendix B); the attribution graph computation (Appendix B.2); and all classifier and baseline configurations (Appendix C.2).

## ACKNOWLEDGMENTS

We would like to thank the anonymous reviewers for their valuable feedback. We are grateful to Edan Toledo and Karen Hambardzumyan for their constructive discussions and insightful feedback on this project. We also thank Shuangrui Ding and Yunzhen Feng for their helpful input, and Megan Ung for assistance with setting up the computing environment.

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

# A    ADDITIONAL DETAILS ON DATASETS

Here we provide a detailed description of our dataset construction, our labeling protocol, and the final dataset statistics.

## A.1    SYNTHETIC DATASET CONSTRUCTION

To create a controlled environment for studying reasoning failures, we procedurally generated two synthetic datasets: **Boolean** and **Arithmetic**. For each, we first generated a ground-truth expression, then prompted our base model (Llama 3.1 8B Instruct) to produce a Chain-of-Thought solution towards solving the expression. We provide the prompt template used to generate CoT in Table 5. Once the CoT is generated, we split them into steps using regular expression.

Table 5: Prompts used for CoT generation across the three datasets. Placeholders for dynamic content are shown in *italics*.

| Dataset | Llama 3.1 8B Instruct Prompt Template |
|---|---|
| **Boolean** | `<|begin_of_text|><|start_header_id|>system<|end_header_id|>` |
| | Evaluate the boolean expression below. |
| | `<|eot_id|><|start_header_id|>user<|end_header_id|>` |
| | {*boolean_expression*} |
| | `<|eot_id|><|start_header_id|>assistant<|end_header_id|>` |
| **Arithmetic** | `<|begin_of_text|><|start_header_id|>system<|end_header_id|>` |
| | Evaluate the arithmetic expression below. |
| | `<|eot_id|><|start_header_id|>user<|end_header_id|>` |
| | {*arithmetic_expression*} |
| | `<|eot_id|><|start_header_id|>assistant<|end_header_id|>` |
| **GSM8K** | `<|begin_of_text|><|start_header_id|>system<|end_header_id|>` |
| | Solve the following math problem step by step. |
| | `<|eot_id|><|start_header_id|>user<|end_header_id|>` |
| | {*gsm8k_question*} |
| | `<|eot_id|><|start_header_id|>assistant<|end_header_id|>` |

**Expression Generation.**    The complexity of the expressions was controlled by the number of operators, $n \in \{3, 5, 7, 10\}$. For the Arithmetic task, we generated expressions with single-digit integers and the operators $\{+, -, \times\}$. For the Boolean task, operands were simply $\{\texttt{True}, \texttt{False}\}$, and possible operators are $\{\texttt{and}, \texttt{or}, \texttt{not}\}$. We generated 10,000 unique expressions for each difficulty level, with the exception of the Boolean task at $n = 3$, where the number of unique expressions is inherently limited. Table 6 shows examples from these datasets with their associated CoTs generated by Llama 3.1 8B Instruct model.

## A.2    LABELING PROTOCOL

Our primary goal was to create labels of the highest possible quality. For the synthetic datasets, we developed a rigorous two-pronged strategy that leverages both a powerful LLM judge and direct programmatic verification, keeping only the annotations where both methods agree.

**Method 1: LLM-as-a-Judge.**    Our first annotation method uses a powerful, external LLM—Llama 3.3 70B Instruct—as an expert judge. For each reasoning step, the judge model was provided with the full problem context and the generated step, and was prompted to output a binary correctness label and a justification. This method was used for all three datasets including the GSM8K dataset. We provide the prompts used for the judge in Table 7 (Boolean), Table 8 (Arithmetic), and Table 9 (GSM8K).

**Method 2: Programmatic State Verification (Synthetic Datasets only).**    For the synthetic tasks, we developed a programmatic method to verify the logical integrity of the entire reasoning trace.

Table 6: Examples of generated expressions and the corresponding CoT generated by our base model for the synthetic datasets.

| Task | Example |
|---|---|
| **Boolean** ($n = 5$) | **Expression:** `(((True or True) and (True and True)) or (True and False))` |
| | **Generated CoT:** To evaluate the boolean expression, we need to follow the order of operations (PEMDAS): 
 1. Evaluate the expressions inside the innermost parentheses: 
   `* (True or True) = True` 
   `* (True and True) = True` 

 2. Now the expression becomes: 
   `* (True and True) or (True and False)` 

 3. Evaluate the expressions inside the parentheses: 
   `* (True and True) = True` 
   `* (True and False) = False` 

 4. Now the expression becomes: 
   `* True or False` 

 5. Evaluate the final expression: 
   `* True or False = True` 

 The final answer is `True`. |
| **Arithmetic** ($n = 7$) | **Expression:** `((((-3)+(-6))*(9*6))+(-4))` |
| | **Generated CoT:** To evaluate this expression, we need to follow the order of operations (PEMDAS): 

 1. Evaluate the expressions inside the innermost parentheses: 
   `* (-3) + (-6) = -9` 

 2. Multiply 9 and 6: 
   `* 9 * 6 = 54` 

 3. Multiply -9 and 54: 
   `* -9 * 54 = -486` 

 4. Add -4 to -486: 
   `* -486 + (-4) = -490` 

 The final answer is -490. |

This goes beyond simply checking the correctness of a single calculation. After each reasoning step $i$ generated by the model, we use a specially crafted prompt to ask the model to output the *current, simplified state* of the original expression.

For example, if the original expression is `(3+5)*2` and the model's first step is `1.  3+5=8`, we then prompt it: "Now the original expression becomes: ". The model is expected to return the reduced expression `8*2`. We then programmatically evaluate the ground-truth value of both the original expression and this new, simplified expression returned by the model. A step is labeled 'correct' only if the two values are identical. If at any point the value of the simplified expression diverges from the ground-truth value of the original, that step is labeled 'incorrect'. While occasionally the model outputs a reduced expression which evaluates to the same value despite being incorrect, this method filters a significant amount of errors.

**Final Label Agreement.** To create our final, high-fidelity label set for the synthetic tasks, we took the intersection of the labels from both methods. That is, a reasoning step was only included in our final dataset if *both* the LLM-as-a-Judge and the programmatic verifier agreed on its label. This strict agreement protocol ensures an exceptionally clean dataset by filtering out ambiguous cases or potential errors from either annotation method.[6]

## A.3 HUMAN VALIDATION OF LLM-AS-A-JUDGE LABELS

To validate the quality of our LLM-as-a-Judge annotation pipeline, a subset of 100 randomly sampled Boolean and Arithmetic expressions ($\approx$700 steps) was independently annotated by four authors. Each annotator labeled half of the set, with every step covered by at least two annotators. To mitigate the rarity of incorrect steps, we upsampled the positive class. Because of the extreme class imbal-

---

[6]While this significantly increases our confidence in the label quality, it also has the effect of making the class distribution more imbalanced, as ambiguous incorrect cases are more likely to be filtered out.

Table 7: Prompt used for step-level annotation by the Llama 3.3 70B Instruct judge model on the Synthetic Boolean dataset. Placeholders for dynamic content are shown in *italics*.

---

**Llama 3.3 70B Instruct Prompt Template**

---

```
<|begin_of_text|><|start_header_id|>system<|end_header_id|>
```

You are an expert in logical reasoning and boolean algebra. You evaluate the correctness of reasoning steps in boolean expression evaluation with high precision.

```
<|eot_id|><|start_header_id|>user<|end_header_id|>
```

Evaluate this reasoning step for logical correctness:

Original Boolean Expression: {*original_expression*}
Correct Truth Value: {*correct_value*}

Context (previous steps):
{*context*}

Step to evaluate: {*step*}

Evaluation criteria:
- Is the boolean operation applied correctly?
- Does the step follow proper order of operations?
- Are the truth values computed accurately?
- Is the reasoning logically sound?

Respond with exactly one of the following:
- CORRECT: if the step is logically sound and mathematically accurate
- INCORRECT: if the step contains logical errors, mathematical mistakes, or invalid reasoning

Your response should start with either "CORRECT" or "INCORRECT" followed by a brief explanation.

```
<|eot_id|><|start_header_id|>assistant<|end_header_id|>
```

---

Table 8: Prompt used for step-level annotation by the Llama 3.3 70B Instruct judge model on the Synthetic Arithmetic dataset. Placeholders for dynamic content are shown in *italics*.

---

**Llama 3.3 70B Instruct Prompt Template**

---

```
<|begin_of_text|><|start_header_id|>system<|end_header_id|>
```

You are an expert in mathematical reasoning and arithmetic operations. You evaluate the correctness of reasoning steps in arithmetic expression evaluation with high precision.

```
<|eot_id|><|start_header_id|>user<|end_header_id|>
```

Evaluate this reasoning step for mathematical correctness:

Original Arithmetic Expression: {*original_expression*}
Correct Value: {*correct_value*}

Context (previous steps):
{*context*}

Step to evaluate: {*step*}

Evaluation criteria:
- Are the arithmetic operations applied correctly?
- Does the step follow proper order of operations (PEMDAS/BODMAS)?
- Are the numerical computations accurate?
- Is the mathematical reasoning sound?

Respond with exactly one of the following:
- CORRECT: if the step is mathematically sound and computationally accurate
- INCORRECT: if the step contains mathematical errors, computational mistakes, or invalid reasoning

Your response should start with either "CORRECT" or "INCORRECT" followed by a brief explanation.

```
<|eot_id|><|start_header_id|>assistant<|end_header_id|>
```

---

ance, Cohen's Kappa ($\kappa$) can underestimate agreement, so we report both $\kappa$ and raw percentage agreement to give a fuller view of inter-annotator reliability.

The results are summarized in Table 10. The agreement among human annotators was moderate as measured by Cohen's Kappa ($\kappa = 0.42$) but high in simple agreement (87.3%). When comparing the consensus human labels to the LLM-as-a-Judge labels, we found fair agreement by Kappa ($\kappa = 0.26$) and similarly high simple agreement (84.1%). A qualitative review of the disagreements revealed a recurring pattern: the vast majority of discrepancies, both among humans and between hu-

Table 9: Prompt used for step-level annotation by the Llama 3.3 70B Instruct judge model on the GSM8K dataset. Placeholders for dynamic content are shown in *italics*.

---

**Llama 3.3 70B Instruct Prompt Template**

```
<|begin_of_text|><|start_header_id|>system<|end_header_id|>
```

You are an expert in mathematical word problems and quantitative reasoning. Your purpose is to evaluate a single reasoning step taken to solve a multi-step word problem. You must be precise, focusing only on the provided step and its relationship to the problem and previously established facts.

```
<|eot_id|><|start_header_id|>user<|end_header_id|>
```

Your task is to evaluate the provided reasoning step for logical and mathematical correctness.

Original Math Problem: {*original_question*}
Correct Final Answer: {*correct_value*}

Context (previous steps):
{*context*}

Step to evaluate: {*step*}

Evaluation criteria:
- Does the step correctly extract and interpret information from the 'Original Problem' or the 'Context'?
- Is it using the right numbers for the right concepts?
- Is the chosen mathematical operation (e.g., addition, subtraction) the correct one to achieve the step's goal, based on the narrative of the 'Original Problem'?
- Is the arithmetic in the step performed correctly?
- Is the mathematical reasoning sound?
- Is the step logically consistent with the problem and previous steps?
- The following types of steps do not contain an error and must be classified as CORRECT:
    - A simple, factually accurate restatement of information from the problem or context.
    - A non-substantive introductory or conversational phrase (e.g., "Let's solve this step by step", "First, we need to find...").

Respond with exactly one of the following:
- CORRECT: if the step is mathematically sound and computationally accurate
- INCORRECT: if the step contains mathematical errors, computational mistakes, or invalid reasoning

Your response should start with either "CORRECT" or "INCORRECT" followed by a brief explanation.

```
<|eot_id|><|start_header_id|>assistant<|end_header_id|>
```

---

mans and the LLM judge, occurred on steps that followed an initial reasoning error. This highlights the inherent ambiguity of labeling steps on a corrupted computational path and directly motivates our strict truncation policy, as detailed in the following section.

Table 10: Inter-Annotator Agreement (IAA) statistics for the human validation study. The comparison shows moderate-to-high agreement, with lower Kappa scores reflecting the extreme class imbalance.

| Comparison | Cohen's Kappa ($\kappa$) | Agreement (%) |
|---|---|---|
| Human vs. Human (Inter-Annotator) | 0.42 | 87.3% |
| Human vs. LLM-as-a-Judge | 0.26 | 84.1% |

## A.4 HANDLING ERROR PROPAGATION

A key challenge in labeling CoT is handling steps that occur after the first mistake. A model can make a technically correct calculation based on a previously incorrect number or expression. This creates a fundamental ambiguity: should a step be judged on its local correctness or its global contribution to the final answer?

To illustrate this problem, consider the flawed CoT trace shown in Table 11. The model correctly evaluates the innermost parentheses in Step 1. However, in Step 2, it makes a critical structural error by incorrectly dropping parentheses, which alters the order of operations. The correct simplification in Step 4 should have been "`not (False or True)`". While subsequent steps like 5 and 7 are arithmetically valid given the new, flawed expression from Step 2, their ultimate correctness is undefined as they exist on a corrupted computational path. To avoid this ambiguity and create a clean, well-defined verification task, we adopted a strict truncation policy. For any given reasoning

Table 11: An example of a flawed CoT trace demonstrating error propagation. The first unrecoverable error in Step 2 renders the correctness of all subsequent steps ambiguous.

---

**Original Expression:** `( not ( ( False or ( True and False ) ) or ( True or False ) ) )`

---

**Model's Generated Chain-of-Thought:** To evaluate this expression, we need to follow the order of operations (PEMDAS):

1. Evaluate the innermost parentheses: `(True and False) = False`. So, `(False or (True and False)) = False or False = False`
2. Now, the expression becomes: `(not (False or (True and False)) or (True or False))`
3. Evaluate the next innermost parentheses: `(True or False) = True`. So, `(not (False or (True and False)) or (True or False)) = (not False or True)`
4. Now, the expression becomes: `(not False or True)`
5. Evaluate the NOT operator: `not False = True`
6. Now, the expression becomes: `True or True`
7. Finally, evaluate the OR operator: `True or True = True`

Therefore, the final result of the boolean expression is: `True`

---

chain, we only retain the steps up to and including the *first* incorrect step. All subsequent steps are discarded from our dataset.

### A.5 DATASET STATISTICS AND MODEL PERFORMANCE

For GSM8K, we used the official testset split. Since its reasoning steps are expressed in natural language and are not always programmatically verifiable, we relied solely on our validated LLM-as-a-Judge pipeline for annotation. Table 12 provides the final statistics for all three datasets, including size and label distribution. Table 13 reports the base performance of our Llama 3.1 8B Instruct model on these tasks. For our synthetic datasets, we randomly split the data into 80% training and 20% testing for the subsequent classification task.

Table 12: Final statistics of our curated datasets, showing the number of reasoning steps and the distribution of correct/incorrect labels after our full annotation and filtering process.

| Dataset | Total Steps | % Correct | % Incorrect |
|---------|-------------|-----------|-------------|
| Synthetic (Boolean) | 126,624 | 99.8% | 0.2% |
| Synthetic (Arithmetic) | 155,434 | 98.8% | 1.2% |
| GSM8K | 8,737 | 93.4% | 6.6% |

Table 13: End-to-end task accuracy of our base model (Llama 3.1 8B Instruct). For the synthetic datasets, we provide a fine-grained breakdown by difficulty, controlled by the number of operators ($n$).

| Dataset | Difficulty (Operators) | Final Answer Accuracy |
|---------|------------------------|-----------------------|
| Synthetic (Boolean) | n=3 | 98.4% |
| | n=5 | 93.27% |
| | n=7 | 89.4% |
| | n=10 | 78.43% |
| Synthetic (Arithmetic) | n=3 | 94.83% |
| | n=5 | 86.8% |
| | n=7 | 73.07% |
| | n=10 | 52.8% |
| GSM8K | - | 75.82% |

## B    TRANSCODER TRAINING DETAILS

Our methodology relies on high-fidelity, sparsely activating transcoders to create an interpretable model. To this end, we trained a TopK-Transcoder for each target MLP module in the Llama 3.1 8B Instruct model. Our training protocol is designed for robustness and follows several best practices established in recent literature.

The transcoders were trained on a high-quality, 10B token subset of the RedPajama-V2 dataset (Weber et al., 2024). We pre-processed the entire training corpus by concatenating and chunking all passages into a uniform length, and we explicitly discarded all beginning-of-sequence (BOS) tokens[7], which we found to be detrimental to stable transcoder training. The transcoder architecture consists of a simple autoencoder with a single hidden layer and a ReLU activation. For each MLP layer in the base model, the transcoder is trained to take the residual stream before the MLP block as input and reconstruct the residual stream after the MLP's computation. The input dimension matches the Llama 3.1 8B's MLP hidden dimension (4096), and the latent feature dimension was set to an overcomplete basis of 131,072. We enforced sparsity structurally using a TopK mechanism, preserving only the $k = 128$ largest feature activations in the forward pass.

We followed several established training techniques to improve feature quality and avoid common pitfalls (Gao et al., 2025; Yang et al., 2025). The decoder weights were normalized to have unit norm, and we did not tie the encoder and decoder weights. To prevent feature collapse, we implemented a dead neuron revival mechanism: if a feature neuron had not activated in 10 million tokens, its activation was forced with an auxiliary loss (coefficient of 1/32).

The transcoders were trained for 4 epochs using the AdamW optimizer. The learning rate was set to 7e-5 with a warmup ratio of 0.5. Training was conducted on 4 nodes, each with 8 Nvidia H200 GPUs, using a total batch size of 4,096. This was achieved with a per-device batch size of 32 and gradient accumulation steps. We found that the training loss generally saturated after approximately 4,000 steps, indicating efficient convergence. We show the training loss on selected layers in Figure 5.

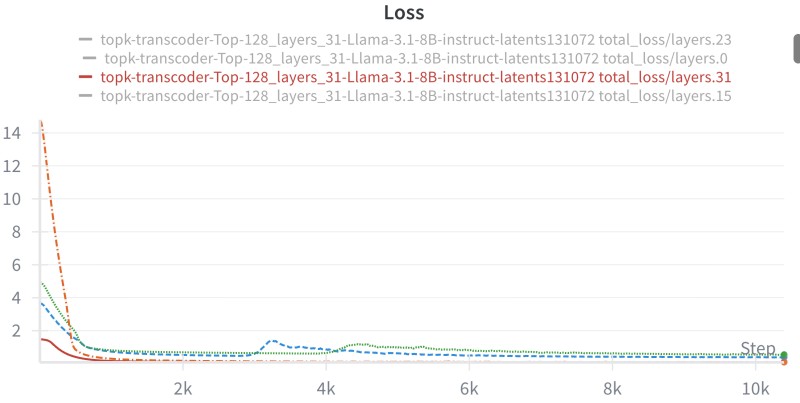

Figure 5: Transcoder Training Loss Curves. The x-axis represents training steps. In all cases, the loss converges efficiently, generally saturating after approximately 4,000 steps.

### B.1    IMPACT OF TRAINING TRANSCODERS ON INSTRUCTION-TUNING DATA

Since our base LLM used is an instruct model, a natural hypothesis is that transcoders fine-tuned on instruction-following data might learn features more relevant to CoT reasoning, thereby improving verification performance. To test this, we trained an version of our transcoders with instruction-tuning (IT) data. Starting from our pre-trained base transcoders, we continued fine-tuning for 1 epoch on the LMSYS-Chat-1M dataset (Zheng et al., 2024), using the same hyperparameters as for

---

[7]BOS tokens are retained when generating activations but their activations are removed afterward for training the transcoders.

Table 14: Performance comparison of CRV with Base transcoders vs. transcoders further trained on Instruction-Tuning (IT) data. Arrows indicate preferred direction (↑ higher is better, ↓ lower is better).

| Transcoder Training | Synthetic (Boolean) | | | Synthetic (Arithmetic) | | | GSM8K | | |
|---|---|---|---|---|---|---|---|---|---|
| | AUROC ↑ | AUPR ↑ | FPR@95 ↓ | AUROC ↑ | AUPR ↑ | FPR@95 ↓ | AUROC ↑ | AUPR ↑ | FPR@95 ↓ |
| Base | 75.87 | 0.97 | 79.17 | 92.47 | 28.92 | 37.09 | 70.17 | 14.3 | 79.61 |
| + IT Data | 76.04 | 1.20 | 66.82 | 91.39 | 28.44 | 38.47 | 72.01 | 15.40 | 83.27 |

the base transcoder training. Following the methodology of Lieberum et al. (2024), we prepended and appended the Llama 3.1 8B Instruct model's IT prefixes to the user queries and model responses respectively.

However, as shown in Table 14, this additional training on IT data did not yield a consistent or meaningful improvement in verification performance on our tasks. This finding is consistent with recent work by Kissane et al. (2024), who found that SAEs trained on base model activations can also faithfully reconstruct the activations of derived IT models. While a deeper mechanistic investigation into how instruction-tuning affects the underlying feature space is a promising direction, we leave this for future work. For our main experiments, we therefore use the more general base transcoders.

## B.2 ATTRIBUTION GRAPH COMPUTATION

**Implementation Details.** We use the implementation from Hanna et al. (2025) to compute attribution graphs. The primary hyperparameters were set as follows: a maximum of 4096 feature nodes, attribution traced from a maximum of 10 logit nodes (selected by a cumulative probability threshold of 0.95), and a batch size of 16 for backward passes. All other parameters follow the repository defaults.

Table 15: Performance comparison of CRV using different token positions for attribution graph computation. The "After" setting computes the graph at the final token of the current step, while "Before" uses the final token of the previous step. Arrows indicate preferred direction (↑ higher is better, ↓ lower is better).

| Attribution Position | Synthetic (Boolean) | | | Synthetic (Arithmetic) | | | GSM8K | | |
|---|---|---|---|---|---|---|---|---|---|
| | AUROC ↑ | AUPR ↑ | FPR@95 ↓ | AUROC ↑ | AUPR ↑ | FPR@95 ↓ | AUROC ↑ | AUPR ↑ | FPR@95 ↓ |
| Before | 68.66 | 1.80 | 77.44 | 85.95 | 12.05 | 47.89 | 70.32 | 16.19 | 85.29 |
| After | 75.87 | 0.97 | 79.17 | 92.47 | 28.92 | 37.09 | 70.17 | 14.3 | 79.61 |

**Ablation on Attribution Position.** The attribution graph is computed with respect to a specific token position. The choice of this position is a critical methodological decision, as it determines which computational moment we analyze. We investigate two hypotheses: analyzing the state before a step is generated (the "pre-computation" trace) versus the state after it is complete (the "post-computation" trace). To test this, we compare two settings: (1) Before: computing the graph at the position of the final token of the previous reasoning step. For the first step of the CoT, this corresponds to the final token of the input question. (2) After: computing the graph at the final token of the current reasoning step, which is the default setting for our main experiments.

The results, presented in Table 15, show a clear and consistent advantage for the "After" setting across nearly all metrics and domains. We hypothesize that this is because the structural signatures of a flawed computation are most fully consolidated in the final token's representation after the step has been fully executed. The pre-computation state may contain signals of intent or planning, but the post-computation state contains the definitive trace of the executed algorithm, including the evidence of its failure. Based on these results, all experiments in the main body of the paper use the "After" (current step) position.

# C    ADDITIONAL CLASSIFICATION DETAILS

## C.1    ATTRIBUTION GRAPH FEATURES

Here we give details about the extracted features for our attribution graphs that we used for our classifier. The feature set is organized into three hierarchical levels:

**1. Global Graph Statistics:** These features provide a high-level summary of the pruned computational graph.

- *Node Counts:* The total number of active transcoder features, as well as the count of transcoder feature nodes and residual stream nodes remaining after pruning. This captures the overall sparsity and composition of the influential subgraph.

- *Logit Statistics:* The probability of the top-ranked token and the entropy of the final logit distribution. These classic uncertainty measures serve as simple but informative baseline features.

**2. Node Influence and Activation Statistics:** This group of features characterizes the properties of the nodes within the pruned graph, moving beyond simple counts.

- *Influence Scores:* The mean influence of all nodes in the pruned graph, along with the total and mean influence specifically from the residual stream ("error") nodes. This helps quantify how much of the final output is attributed to specific learned features versus the model's direct pass-through states.

- *Activation Statistics:* For the pruned transcoder feature nodes, we compute the mean, max, and standard deviation of their activation values. This captures the intensity and distribution of the active, interpretable features. A high maximum activation, for instance, might signal that a single, highly decisive feature was responsible for the step.

- *Layer-wise Feature Histogram:* A histogram of active transcoder features across the model's layers. This feature vector characterizes the distribution of computational effort across the model's depth, allowing us to test hypotheses such as whether errors correlate with the activation of components at specific layers.

**3. Topological and Path-Based Features:** To capture the structure and efficiency of the information flow, we compute a rich set of topological features on the pruned, directed subgraph.

- *Edge and Density Statistics:* Aggregate statistics on the edge weights (sum, mean, std), the total number of edges, and the graph density. We hypothesize that a sparse, fragmented graph (low density, few edges) may indicate a breakdown in information flow characteristic of an error.

- *Centrality Measures:* To identify critical "hub" nodes in the computation, we calculate the mean and max for both degree centrality and weighted betweenness centrality. These features assess whether influence is concentrated or diffused.

- *Connectivity and Path Lengths:* The number of weakly connected components and the average shortest path length within the largest component. A highly fragmented graph may suggest a failed computation. A particularly crucial feature is the **shortest path length from any input token node to any final logit node**. This directly measures how efficiently information from the prompt propagates to the final decision. A long or non-existent path is hypothesized to be a strong signal that the model is "ignoring" its instructions or context.

## C.2    ADDITIONAL DETAILS ON BASELINES

Here we provide additional implementation details for the baseline methods used in our main experiments, ensuring full reproducibility.

**Black-Box Baselines.**    This category includes methods that operate solely on the output logits of the final token for each reasoning step. We use implementations from Wang et al. (2025a).

**Gray-Box Baselines.** This category includes methods that leverage the model's internal hidden states. For **CoE** (Wang et al., 2025a) and **CoT-Kinetics** (Bi et al., 2025), which are training-free, we followed the official implementations and protocols described by their respective authors to compute the verification scores. We set $\gamma$ in CoT-Kinetics to 0.8, and use mean pooling for reasoning token aggregation.

For our supervised **LR Probe** baseline, the choice of which layer's hidden states to use is a hyper-parameter. To determine the optimal layer for each dataset, we performed a hyperparameter search, training a separate probe on the average hidden states from each of the 32 layers of Llama 3.1 8B Instruct on a small validation split. This process allowed us to identify the layer that contained the most predictive signal for each distinct reasoning task. The best-performing layers, which were subsequently used for the main results reported in Table 1, were found to be:

- **Layer 0** (the token embedding layer) for the Synthetic (Boolean) dataset.
- **Layer 9** for the Synthetic (Arithmetic) dataset.
- **Layer 0** (the token embedding layer) for the GSM8K dataset.

Similar to the protocol for the LR probe, we trained separate **MLP Probes** on the hidden states of specific layers. We searched over a hyperparameter grid similar to Zhang et al. (2025), varying the learning rate, weight decay, hidden dimension size ($d$), and the source layer index. The complete search space is provided in Table 16. We selected the best hyperparameters for each dataset based on a small validation set. The optimal configurations are reported in Table 17.

Table 16: Hyperparameter search space for the MLP Probe baseline.

| Hyperparameter | Values Searched |
|---|---|
| Source Layer Index | $\{0, 1, \ldots, 31\}$ |
| Hidden Dimension ($d$) | $\{16, 32, 64, 128\}$ |
| Learning Rate | $\{1\text{e-}3, 1\text{e-}4, 1\text{e-}5\}$ |
| Weight Decay | $\{1\text{e-}1, 1\text{e-}2, 1\text{e-}3, 1\text{e-}4, 1\text{e-}5\}$ |

Table 17: Optimal hyperparameters for the MLP Probe on each dataset. The Layer column indicates the hidden state layer that yielded the highest verification performance.

| Dataset | Layer | Learning Rate | Weight Decay | Hidden Dim ($d$) |
|---|---|---|---|---|
| Synthetic (Boolean) | 0 | 1e-3 | 1e-1 | 16 |
| Synthetic (Arithmetic) | 12 | 1e-5 | 1e-1 | 16 |
| GSM8K | 0 | 1e-4 | 1e-5 | 128 |

### C.3 ADDITIONAL DETAILS ON EVALUATION METRICS

We report three evaluation metrics: the Area Under the Receiver Operating Characteristic curve (AUROC), the False Positive Rate at 95% True Positive Rate (FPR@95), and the Area Under the Precision–Recall curve (AUPR). For all metrics, we treat the verifier output as a real-valued score $s(x) \in \mathbb{R}$, where higher values indicate a higher predicted likelihood of an error. The ground-truth labels are $y \in \{0, 1\}$, where $y = 1$ denotes an incorrect reasoning step (the positive class) and $y = 0$ denotes a correct step.

**AUROC.** The Area Under the ROC Curve is defined as the integral of the True Positive Rate (TPR) against the False Positive Rate (FPR):

$$\text{AUROC} = \int_0^1 \text{TPR}(\tau) \, d(\text{FPR}(\tau)).$$

The rates at a given threshold $\tau$ are defined as:

$$\text{TPR}(\tau) = \frac{\sum_x \mathbf{1}[s(x) \geq \tau \wedge y = 1]}{\sum_x \mathbf{1}[y = 1]}, \quad \text{FPR}(\tau) = \frac{\sum_x \mathbf{1}[s(x) \geq \tau \wedge y = 0]}{\sum_x \mathbf{1}[y = 0]}.$$

We compute this via the trapezoidal rule by sweeping $\tau$ across all observed score values.

**FPR@95.** This metric captures the False Positive Rate at a specific operating point where the recall (TPR) is fixed at 0.95. It is defined as:

$$\text{FPR@95} = \text{FPR}(\tau_{0.95}),$$

where $\tau_{0.95}$ is the threshold such that $\text{TPR}(\tau_{0.95}) = 0.95$. In practice, we select the strict threshold $\tau$ that yields a TPR closest to, but not less than, 0.95.

**AUPR.** The Area Under the Precision-Recall Curve quantifies the trade-off between precision and recall for the positive (error) class. It is defined as:

$$\text{AUPR} = \int_0^1 \text{Precision}(\text{Recall})\, d(\text{Recall}),$$

where Precision at threshold $\tau$ is:

$$\text{Precision}(\tau) = \frac{\sum_x \mathbf{1}[s(x) \geq \tau \wedge y = 1]}{\sum_x \mathbf{1}[s(x) \geq \tau]}.$$

All metrics were computed using the standard implementations provided by the scikit-learn library (Pedregosa et al., 2011), applied to the held-out test sets.

## C.4 ADDITIONAL CLASSIFIER AND THEIR RESULTS

To validate our choice of a Gradient Boosting classifier for the main experiments, we benchmarked its performance against several standard alternatives on our curated graph feature set. We evaluated a simple baseline, a linear model, and another tree-based ensemble to understand the trade-offs between model complexity and verification performance. For this analysis and main experiments in this work, we used the default hyperparameters from the scikit-learn library for each classifier, as an initial, non-exhaustive hyperparameter search did not yield any significant improvements, suggesting that the feature set itself provides a strong signal that is not overly sensitive to classifier configuration.

Table 18: Performance comparison of different diagnostic classifiers. Arrows indicate preferred direction ($\uparrow$ higher is better, $\downarrow$ lower is better).

| Method | Synthetic (Boolean) | | | Synthetic (Arithmetic) | | | GSM8K | | |
|---|---|---|---|---|---|---|---|---|---|
| | AUROC $\uparrow$ | AUPR $\uparrow$ | FPR@95 $\downarrow$ | AUROC $\uparrow$ | AUPR $\uparrow$ | FPR@95 $\downarrow$ | AUROC $\uparrow$ | AUPR $\uparrow$ | FPR@95 $\downarrow$ |
| Dummy | 50.8 | 0.25 | 100 | 49.84 | 1.20 | 100 | 48.06 | 6.46 | 100 |
| Logistic Regression | 76.4 | 0.75 | 68.91 | 89.5 | 11.46 | 41.56 | 73.8 | 18.70 | 78.69 |
| Random Forest | 61.71 | 4.49 | 100 | 92.99 | 43.68 | 30.56 | 71.7 | 17.65 | 76.18 |
| Gradient Boosting | 75.87 | 0.97 | 79.17 | 92.47 | 28.92 | 37.09 | 70.17 | 14.3 | 79.61 |

The results are presented in Table 18. As expected, the Dummy classifier, which makes predictions based on the training set's class distribution, performs near chance level (AUROC $\approx$ 50). This confirms that our graph features contain a significant predictive signal that is non-trivial to learn. Interestingly, a standard Logistic Regression model achieves competitive performance, yielding the best AUROC on two of the three datasets and the strongest overall results on GSM8K. This indicates that the features are highly informative even with a simple linear model.

However, the tree-based ensembles often achieve superior performance on other key metrics. The Random Forest classifier, for instance, yields a substantially higher AUPR and lower FPR@95 on the complex Arithmetic dataset, suggesting its ability to capture non-linear feature interactions is critical for high-precision verification in that domain. Overall, no single classifier is dominant across all domains and metrics. We chose Gradient Boosting for our main experiments as it consistently provides a strong and robust performance profile, but these results highlight that the optimal choice of diagnostic classifier may be domain-specific.

## C.5 ADDITIONAL RESULTS FOR RQS

Here we provide additional results for our research questions. We first show distributions of highly predictive features for correct versus incorrect reasoning steps on our synthetic datasets (Figure 6

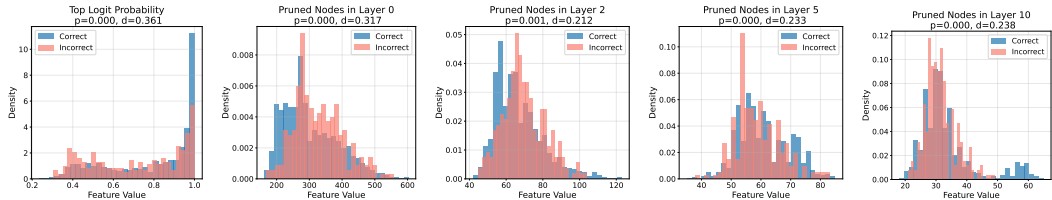

Figure 6: **Topological Fingerprints of Error on Arithmetic.** Distributions of five selected graph features for correct (blue) vs. incorrect (red) reasoning steps. The visual separation is statistically significant for each feature shown (independent t-test, $p < 0.001$) and represents a medium-to-large effect size (Cohen's d). This provides quantitative evidence that attribution graphs contain a clear, separable structural signal of a computation's integrity.

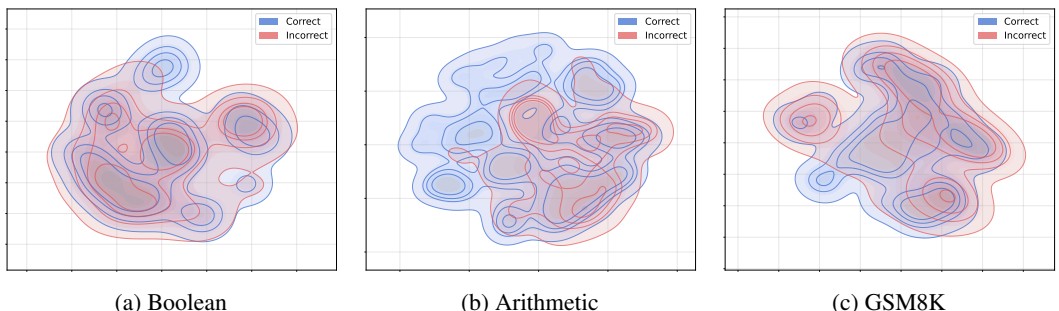

Figure 7: **Topological Fingerprints of Error on Boolean.** Distributions of five selected graph features for correct (blue) vs. incorrect (red) reasoning steps. The visual separation is statistically significant for each feature shown (independent t-test, $p < 0.001$) and represents a medium-to-large effect size (Cohen's d). This provides quantitative evidence that attribution graphs contain a clear, separable structural signal of a computation's integrity.

|  |  |  |
|:---:|:---:|:---:|
| (a) Boolean | (b) Arithmetic | (c) GSM8K |

Figure 8: Distributions of features after t-SNE for correct (blue) vs. incorrect (red) reasoning steps. The separation validates that CRV captures a distinct structural fingerprint of error.

for arithmetic; Figure 7 for Boolean). Next, we display the distributions of full feature vectors after t-SNE projection in Figure 8.

We demonstrate another casual intervention with a concrete case study on the arithmetic task, where we correct a subtle mathematical parsing error not by suppressing a faulty feature, but by *amplifying* a correct one. We present the model with the expression (- (5+ (4*9) ) ). As shown in Table 19, the model initially misinterprets the leading unary minus, treating it as a subtraction from an implicit -1 and producing the incorrect result of -42. Our CRV classifier flagged this step as incorrect, and its feature importance analysis suggested that the error stemmed from the absence of a strong "multiplication" signal.

Hypothesizing that the model failed to recognize the unary minus as a multiplication by -1, we performed a direct intervention. We identified a transcoder feature from the graph whose semantics corresponded to multiplication and, using a forward hook, manually amplified its activation value just before the model generated its erroneous step. The effect was immediate and successful. As shown in the "After Intervention" column, this boost was sufficient to steer the model onto the correct computational path. It now correctly multiplies by -1 and arrives at the final answer of -41.

Table 19: A successful causal intervention via feature amplification. A subtraction error is corrected by amplifying a multiplication feature, steering the model to the correct reasoning path.

| Before Intervention (Incorrect) | After Intervention (Correct) |
|---|---|
| Evaluate the arithmetic expression below. ( - ( 5 + ( 4 * 9 ) ) ) To evaluate this expression, we need to follow the order of operations (PEMDAS): 1. Multiply 4 and 9: 4 * 9 = 36 2. Add 5 and 36: 5 + 36 = 41 3. Subtract 41 from -1: -1 - 41 = -42 The final answer is -42. | Evaluate the arithmetic expression below. ( - ( 5 + ( 4 * 9 ) ) ) To evaluate this expression, we need to follow the order of operations (PEMDAS): 1. Multiply 4 and 9: 4 * 9 = 36 2. Add 5 and 36: 5 + 36 = 41 3. Multiply -1 by 41: -1 * 41 = -41 The final answer is: -41. |

## D   USAGE OF LANGUAGE LANGUAGE MODELS

In the preparation of this manuscript, we utilized LLMs as writing assistants. Their role was strictly limited to improving the clarity, conciseness, and grammatical correctness of the text. The authors take full responsibility for all content and any remaining errors.

## E   LIMITATIONS

Our work introduces a new methodology for the scientific analysis of reasoning, and its limitations are intrinsically linked to its design as a white-box, mechanistic instrument.

**Computational Intensity.**   A primary limitation of CRV is its computational intensity. The process, which involves training a suite of transcoders, replacing model modules, and constructing a detailed attribution graph for every reasoning step, is orders of magnitude more resource-intensive than black-box or gray-box verification methods. This cost is a direct consequence of our white-box approach, which prioritizes mechanistic depth over practical efficiency. As such, CRV in its current form is positioned as a scientific tool for deep analysis, not as a scalable, real-time verifier for production systems.

**Aggregative vs. Feature-Level Analysis.**   The feature set used by CRV is primarily aggregative; it captures statistical and topological properties of the graph, such as node counts, influence scores, and density. As an early work, it does not yet fully exploit the semantic content of the individual transcoder features that constitute the graph's nodes. For instance, our current classifier learns statistical correlations over the entire feature set; it does not reason symbolically about whether a specific feature for numerical addition is appropriately activated by numerical inputs. This represents a significant opportunity. A promising future direction lies in developing more sophisticated classifiers or rule-based systems that operate directly on the semantics of these disentangled features, paving the way for a new class of neuro-symbolic verifiers.

**Generalizability of Error Signatures.**   Our empirical results are based on a single model family (Llama 3.1) at the 8B scale. Whether the precise structural fingerprints we identified generalize to different architectural paradigms, such as Mixture-of-Experts, or across significant model scales (e.g., 70B and larger) remains an open question. Furthermore, as our cross-domain experiments revealed, the error signatures are highly domain-specific. Our work provides a strong foundation and a methodology for discovering these signatures, but further studies are needed to determine if more universal principles of computational failure exist.

**Fidelity of Interpretability Tools.**   The validity of our analysis is contingent on the quality and fidelity of the underlying interpretability tools. The features identified by our transcoders, while demonstrably useful, represent one possible sparse basis and are not exhaustive. Similarly, the attribution method provides a powerful but ultimately incomplete approximation of the true information flow within the model. Future improvements in these foundational techniques, such as the develop-

ment of more faithful sparse autoencoders or more precise attribution methods, will directly enhance the resolution and reliability of analyses like ours.

