# OpenReview forum: "Verifying Chain-of-Thought Reasoning via Its Computational Graph"
_ICLR.cc/2026/Conference — ICLR 2026 Oral_

### Official Review · Reviewer_Mmoa · 2025-10-28

**Soundness:** 3
**Presentation:** 3
**Contribution:** 3
**Rating:** 6
**Confidence:** 5

**Summary:**

The authors leverage recent mechanistic interpretability techniques to develop a new white-box method for assessing the internal computations performed by a model while solving simple reasoning tasks. They show that they, using their circuit-based reasoning verification (CRV) outperform considered greybox and blackbox baselines on CoT-step-correctness prediction tasks on three small synthetic and math datasets.

**Strengths:**

Advancing the field of mechanistic interpretability towards the practically relevant task of model debugging is a timely and highly relevant contribution. This work makes an important first step towards a scalable "white-box" technique for systematically tracing back model failures (in this case reasoning errors) to models' internal states and (hypothesized) learned algorithms.

The work performs thorough evaluations with several decent baselines and including several interesting ablations to understand the strengths and weaknesses of the proposed method. E.g., OOD evaluation shows that the method does not transfer well to unseen reasoning tasks and the feature analysis shows that ablating the topological graph features hurts the performance least. While in particular the second finding is a little bit in tension with the storyline of the paper of exploiting structural properties of the models latent algorithms, it is good scientific practice.

It is interesting that a mostly straightforward adaptation of transcoder based circuits in the proposed nice framing of graph classification outperforms the considered baselines on partial-CoT-correctness prediction.

The qualitative causal experiment of building CRV-informed interventions to correct LLM outputs shows that methods like this may unlock unexpected downstream usecases.

**Weaknesses:**

My major concerns with this paper are:
* In the main experiment in table 1 the probing baseline gets outperformed by most of the black-box methods. This suggests to me that the probing setup might be flawed: the datasets seem tiny, was there enough data to train the probes? Probing performance is sensitive to the layer and token position. While it is common to aggregate hidden states over multiple tokens, if token sequences get too long like in CoT this seems like a too simplistic way of probe construction. Maybe it would make sense to consider a simple 1-layer attention based aggregation of the hidden states instead of the mean or a sliding window or something like that. I would be surprised if probing really underperforms simple baselines like the black-box ones as suggested by table 1.
* The authors emphasize that the white-box nature of their approach is important to achieve this level of performance and being able to pick up on structural graph properties were the key factor for the success of the method. However, they also show that gradient boosting based classifier is mostly robust to ablation of topological feature and achieves almost the same level of performance considering only local and global graph features. This to me also suggests that a much simpler grey-box technique, e.g., not requiring a substitute, transcoder-based attribution should achieve similar performance on the considered tasks.
* The approach is termed white-box however its white-box aspects are under-explored. The qualitative CRV-inspired intervention goes into the right direction, however, an in-depth analysis of the white-box aspects of the work would significantly strengthen the contribution. Another thing to consider to add towards the whitebox/interpretability/understanding framing would be to leverage transcoder feature annotations based on max activating examples and so on to help the debugging process.

Minor concerns:
* Many things, such as explanation of the circuit computation and baseline methods, are deferred to the related work. It would be nice to convey the main ideas and intuitions in the main paper. If the paper gets accepted I'd like to request a short explanation of the attribution based circuit finding algorithm and some of the baselines considered in table 1.
* Some related work on reasoning model interpretability that is closely related is missing in the related work section. E.g., [1] provide some preliminary evidence that CoT reasoning indeed involves some learned algorithms. The same goes for [2], which reverse engineers a self-verification circuit in a toy-model.

[1] https://arxiv.org/abs/2510.07364v3
[2] https://arxiv.org/abs/2504.14379

**Questions:**

My overall assessment of the paper would improve if the authors convincingly address the main concerns raised in the weaknesses section.

I would also encourage the authors to provide stronger evidence supporting their claim that the proposed approach is genuinely white-box in nature.

---

> ### Author Response · Authors · 2025-11-20
> **Author's rebuttal (part 1)**
>
> We thank the reviewer for the constructive comments. Please find our responses to your comments below on the major concerns.
>
> _**Concern 1**_
>
> We shared your initial surprise that black-box methods outperformed the probe, as this contradicts common intuition in other domains (like truthfulness). However, we believe this result is robust and scientifically meaningful, rather than an artifact of the setup.
> - Regarding data size and tuning: While GSM8K is smaller, our Synthetic datasets contain over 120,000 examples, yet the probe exhibits the same performance gap there. This rules out data scarcity as the root cause. Furthermore, as noted in Appendix C.2, we performed an exhaustive search over all 32 layers and multiple hyperparameters to ensure the probe had every advantage.
> - Regarding aggregation method: Regarding the concern that mean-pooling "smears" the signal: unlike full documents, individual CoT reasoning steps are typically short (often a single sentence or equation). In our experiments, mean-pooling outperformed last-token pooling. While attention-based aggregation is a valid suggestion, the failure of the probe even on short, simple synthetic steps suggests the issue is not how we aggregate the state, but the nature of the state itself.
> - Regarding the new MLP baseline: To definitively test if the limitation was the linear constraint (or simple aggregation), we added a non-linear MLP Probe baseline in the revision (please check the updated pdf). Even with this increased capacity, the probe failed to close the gap with CRV.
>
> This reinforces our core finding: reasoning errors (unlike factual hallucinations) do not always leave a linearly separable "error mark" in the static hidden state. Often, the computation is flawed while the representation remains plausible. This is exactly why a structural method like CRV is necessary.
>
> _**Concern 2**_
>
> You are correct that removing the “Topological and Path-Based Features” group produces the smallest drop in performance. However, it is important to note that the most predictive feature group, “Node Influence and Activation Stats”, is also rooted in the graph. These values are not simple activation statistics. A node’s influence measures its causal effect on the final output, and this can only be computed by following paths in the attribution graph. In addition, the activation statistics in this group are taken only over the pruned subgraph of causally relevant nodes, rather than over all active features.
>
> Thus, the ablation study does not imply that the graph is unimportant. Instead, it shows a more detailed picture: for these tasks, local graph properties, such as the influence of particular nodes, carry more predictive signal than global properties such as overall graph density or connectivity. Both kinds of information come from the graph.
>
> This also explains why a simpler gray-box method would likely struggle. A probe trained on raw activations, even sparse ones, would know only which features are active. It would not know how these features affect the final computation of the step. The attribution graph supplies the causal and relational structure, and our results indicate that this structure provides the strongest signal for identifying reasoning failures.
>
> _**Concern 3**_
>
> The central aim of this work is to introduce the CRV methodology, show that it provides reliable verification, and demonstrate a causal link through a controlled intervention. We view this as a foundational step. Your point about using the full white-box representation for deeper analysis aligns well with our next step. The idea of using maximally activating examples to assign semantic meaning to transcoder features is fully consistent with how we expect this methodology to evolve.
>
> In our manual case study (Table 4), we proposed that one feature was associated with “multiplication”. A more systematic version of this process would follow the outline you describe:
> 1. automatically detect the anomalous feature using the classifier’s importance scores, and
> 2. consult a pre-computed library to identify inputs that maximally activate this feature, such as phrases involving “the product of X and Y”, “times”, or “multiplied by”.
> 3. This automated semantic labeling would give the developer a clear and immediate hypothesis about the failure mode, such as “the model is activating a multiplication-related feature at the wrong stage”, which would greatly speed up debugging.
>
> This deeper semantic analysis is a substantial research direction on its own. Although a full system is outside the scope of this initial paper, we appreciate the suggestion and included an explicit discussion of this future direction in Section 4.4, emphasizing that it is a core benefit of the CRV framework.

---

> ### Author Response · Authors · 2025-11-20
> **Author's rebuttal (part 2)**
>
> Please fine below our response on the minor concerns.
>
> Thank you for the suggestions. We currently provide a high-level overview of the circuit finding algorithm in Section 3.2.2 and list the baseline categories in Section 4.1. We also thank the reviewer for pointing us to these related works. We agree that both papers are relevant to the mechanistic understanding of CoT and have incorporated them into our revised Related Work section. We would like to note, however, that [1] was first released on arXiv on October 8, 2025. As this date falls after the ICLR submission deadline, we consider it concurrent work.
>
> [1] https://arxiv.org/abs/2510.07364v3

---

> > ### Comment · Reviewer_Mmoa · 2025-11-24
> > **Acknowledgment of response**
> >
> > I acknowledge the author's response.
> >
> > Most of my concerns have been addressed, except that I would like to see how a circuit agnostic deep-sequence-modelling-based baseline (transformer or linear attention architecture) would perform in this setting. However, I think this might be out of the scope of the current paper, which already has strong contributions such as highlighting CRV as an interesting novel area of research.
> >
> > I raise my score.

---

> > > ### Author Response · Authors · 2025-11-24
> > > **Reply to reviewer's response**
> > >
> > > We sincerely thank the reviewer for their engagement and for raising their score. We agree that a Transformer-like baseline is an excellent suggestion for future work. We also appreciate your recognition of CRV as a novel research direction.

---

### Official Review · Reviewer_FZ7t · 2025-10-28

**Soundness:** 3
**Presentation:** 3
**Contribution:** 3
**Rating:** 8
**Confidence:** 4

**Summary:**

The paper demonstrates a great use case of using interpretability techniques for a practical application of detecting invalid cases of chain of thought. In particular, the authors build out an attribution graphs for valid and invalid CoT steps. These graphs in turn can be turned into features for a classifier to classify valid and invalid CoT steps, using graphical features such as global graph statistics, node statistics, or topological features. These features are used to train a Gradient Boosting Classifier.

These experiments were conducted on carefully crafted datasets, for a boolean reasoning task, arithmetics, and carefully annotated GSM8k. The authors compare to a wide range of baselines, each of which are categorized as "black-box" or "gray-box" methods, in which their approach (CSV) outperforms all baselines.

I appreciate that the authors check for cross-domain generalization of their CRV classifier. It turns out that CRV does not generalize well, suggesting that CRV (and the underlying attribution graph) likely uses specific signatures per domain. It would be interesting to see how to build a CRV model that generalizes well across domains.

The authors also demonstrate that CRV's stay effective regardless of the difficulty of the problem. The authors control for task difficulty with number of steps required to solve the problem, and CRV remains effective as task difficulty is scaled.

A natural question that arises is what kind of features CRV relies on. By ablating away some features from the attribution graph, the authors identify that "local features" (i.e., node influence) matter the most, while "global features" such as topological features matter less.

The authors dig deeper to look at transcoder features (as opposed to features of the attribute graph) to identify specific concepts. As expected and as seen in numerous SAE work, intervening on these features have causal power to control the CoT reasoning traces.

Overall a strong contribution that leverages attribution graphs for a tangible application.

**Strengths:**

Overall the paper is nicely scoped and demonstrates a clear win in leveraging attribution graphs to identify valid and invalid chain-of-thought steps. This provides a practical benefit of recent interpretability methods, which (as exciting as interpretability findings are) don't always happen. I hope to see more work that connects from interpretability insights to tangible actionable applications.

**Weaknesses:**

Overall the paper make claims/contributions that are well-scoped and nicely backed by their experiments. If I had to list out some weaknesses....

* Figure 3: How much of this visualization is an artifact of projecting high-dimensional features to 2d? The claims of "computational near misses" and "zone of computational integrity" seem a bit hand-wavy based on these visualizations.

* I'm not sure what other baselines would have been possible, but IIUC some of these baselines seem a bit unfair in that some of them don't have as much access to the LM's hidden states? For instance, PPL/MaxProb/PPL are overly simplified scalar summarizations of the model's forward pass. Is that correct?

* Any thoughts on how to scale the diagnoses of identifying relevant features (of the transcoder) and trying causal intervention experiments?

* I appreciate that the authors tested for cross-domain generalization, although the results seem to indicate that they don't. Any thoughts on how to build a diagnostic classifier that does generalize across domains?

* The fact that LR probe performed so poorly is a bit surprising. It's also very surprising to see that layer 0 worked best for the LR probe. Why do you think Layer 0 performed best? Unrelatedly, could the poor performance of LR probe be cause of averaging the hidden states across tokens? Lastly, do you think simply using mean differences of hidden states (for correct vs. incorrect cases) could work better?

**Questions:**

Most of my questions have been asked in the "weaknesses" section.

---

> ### Author Response · Authors · 2025-11-20
> **Author's rebuttal (part 1)**
>
> We thank the reviewer for their insightful feedback. Please find our response to the questions below.
>
> _**Weakness 1**_
>
> We agree that projections must be interpreted cautiously. The visualizations in Figure 3 are intended to provide intuition for patterns that are firmly established by our quantitative results. First, the high AUROC scores (e.g., 92.47 on Arithmetic) prove that the classes are mathematically separable in the full feature space. Second, the histograms in Figure 4 demonstrate that individual structural features exhibit statistically significant separation. If anything, Figure 3 and 4 don’t convey the full extent to which correct and incorrect steps become separable in the full feature space of the classifier. Therefore, the "zone of integrity" is not a visual artifact but a reflection of the classifier's decision boundary: correct steps form a tight structural cluster, while incorrect steps often lie on the periphery, sharing general properties with valid reasoning but lacking specific markers of correctness.
>
>
> _**Weakness 2**_
>
> We divided our baselines into two categories to reflect the different levels of access to the model:
> - Black-box baselines (MaxProb, PPL, Entropy, and related methods) use simple scalar summaries of the final logit distribution. We included them because they are standard, widely used, and trivial to compute for uncertainty estimation. They serve as a natural starting point and form a necessary reference for any stronger verifier.
> - Gray-box baselines (CoE, CoT-Kinetics, LR Probe) use the model’s internal hidden states and therefore offer a closer point of comparison to CRV. This group represents the baselines that operate with the same level of access as our method. In the updated version of the paper, we have also added a new MLP probe, a nonlinear and trainable model applied to the hidden states.
>
> The main result is that CRV outperforms these gray box methods as well. For example, on the Arithmetic task, CRV achieves an AUROC of 92.47 compared to 69.39 for the strongest gray box baseline. This confirms that our performance gain stems from analyzing the causal graph structure rather than simply having access to internal states.
>
> _**Weakness 3**_
>
> This is an important point and it highlights a central direction for future research. Our current case study is manual and time-consuming, meant only to show that causal intervention is feasible. Scaling this process is challenging but also promising. We see a path toward a semi-automated pipeline with the following stages:
> - After training the diagnostic classifier, we can automatically identify which features are most predictive of failure by using feature importance scores from the model. This would surface signals like “high activation of feature X in layer Y”, which may be tied to a recurring error pattern.
> - The next stage is to assign a human-interpretable meaning to “feature X”. This remains the most difficult step. Current work in mechanistic interpretability often uses searches for inputs that strongly activate a given feature to infer its likely function. Extending such methods to the transcoder setting would be essential for automation.
> - With a set of implicated features and tentative semantic descriptions, a researcher could form hypotheses much more efficiently. For example, one might suspect that the model is triggering a multiplication-related feature at the wrong point in the computation and could then design a targeted intervention to test this idea.
>
> A completely automated pipeline will require significant advances, but a semi-automated interpretability interface of this kind could greatly accelerate the analysis and correction of model failures.

---

> ### Author Response · Authors · 2025-11-20
> **Author's rebuttal (part 2)**
>
> _**Weakness 4**_
>
> We view this lack of generalization as an important insight from our work. It is not a limitation of the CRV method, but rather an observation about how LLMs reason. Errors from different domains, such as arithmetic and logic, arise from distinct computational patterns.
>
> Building a classifier that generalizes across domains is an important next step. We see two possible paths forward.
>
> - One option is to train a single diagnostic classifier on a large dataset that mixes multiple domains. In response to your comment, we ran this experiment (added as CRV (Combined) in Table 2 in the revised paper). The results confirm that this path is viable: the combined model recovers part of the performance gap observed in the zero-shot setting. This proves that while error signatures are distinct, they are not mutually exclusive; a sufficiently broad classifier can learn the "union" of these failure modes. A more sophisticated version could include embedding tasks in a common space where similarity corresponds to similarity in correctness/incorrectness indicators, and then condition the classifier on the embedding of the task under consideration.
>
> | Test Set  | Method (Train Set)      | AUROC ↑ | AUPR ↑ | FPR@95 ↓ |
> |-----------|--------------------------|---------|--------|----------|
> | Boolean   | Baseline (MaxProb)       | 58.81   | 0.34   | 95.20    |
> |           | CRV (GSM8K)              | 45.77   | 0.21   | 97.28    |
> |           | CRV (Arithmetic)     | 61.58 | 0.51 | 87.55 |
> |           | CRV (Combined)           | 65.18   | 0.65   | 80.15    |
> |           | *CRV (Boolean)*          | *75.87* | *0.97* | *79.17*  |
> | Arithmetic| Baseline (Energy)        | 76.45 | 5.59 | 73.86 |
> |           | CRV (GSM8K)              | 55.11   | 1.50   | 91.91    |
> |           | CRV (Boolean)            | 69.59   | 2.64   | 72.87 |
> |           | CRV (Combined)           | 90.51   | 30.57  | 44.23    |
> |           | *CRV (Arithmetic)*       | *92.47* | *28.92* | *37.09* |
> | GSM8K     | Baseline (Energy)        | 62.55 | 9.11 | 86.34 |
> |           | CRV (Boolean)            | 44.37   | 6.33   | 95.71    |
> |           | CRV (Arithmetic)         | 57.04   | 7.85   | 94.37    |
> |           | CRV (Combined)           | 70.62   | 12.66  | 80.22    |
> |           | *CRV (GSM8K)*            | *70.17* | *14.3* | *79.61* |
>
> - A second, more ambitious path is to look for lower-level structural markers of failure that do not depend on any specific task. For instance, errors across domains might share early signs such as disrupted information flow between layers, a fragmented attribution graph, or a failure to attend to important input tokens. By training the classifier to focus on these shared structural properties, it may be possible to learn a more general model of computational failure.
>
> _**Weakness 5**_
>
> We have several hypotheses that could explain the pattern you observed.
>
> For Layer 0, our main hypothesis is that, for a simple linear probe, the most distinguishable signal in tasks such as GSM8K and Boolean logic appears in the initial embedding space. Layer 0 encodes the input tokens directly. The classifier could be learning that the model finds harder inputs with a given distribution of tokens (e.g. many multiplications, or logical negations). Some errors may arise from an early misunderstanding of symbols or numbers, and this type of signal is clearest at the embedding layer. As the representation moves through the network, it becomes more abstract and more shaped by non-linear transformations. A linear classifier may not be able to recover the relevant structure at those later layers.
>
> Regarding the effect of averaging, this is a reasonable concern. In footnote 3, we report that we also tested a last-token probe. Averaging the hidden states across all tokens in the generated step produced slightly higher and more stable performance than using only the last token. This indicates that averaging is not the main factor behind the lower performance.
>
> Your suggestion about using mean differences is insightful. A classifier based on mean differences of hidden states belongs to the same class of linear probes. Since our LR probe already shows difficulty finding a strong linear boundary, a simpler linear method based on mean differences would likely face the same issue. The signal that distinguishes correct and incorrect steps in dense hidden states seems weak, which is exactly the limitation that our white-box method is designed to address.

---

> > ### Comment · Reviewer_FZ7t · 2025-11-26
> >
> > Thank you for your response and additional experiments.
> >
> > I have read through all other discussions and have no other concerns. The concerns of reviewer 9q4r are valid, and I am also surprised to see such low scores for the linear probe (which I also mention in my comments) but the authors seem to have done a thorough experiment to double check this point.
> >
> > With that being said, I will maintain my score.

---

> > > ### Author Response · Authors · 2025-11-27
> > > **Reply to Reviewer's Response**
> > >
> > > Thank you for your time and for considering the parallel discussions. We are glad that the added experiments and the other responses addressed your concerns.

---

### Official Review · Reviewer_9q4r · 2025-10-31

**Soundness:** 2
**Presentation:** 3
**Contribution:** 3
**Rating:** 4
**Confidence:** 4

**Summary:**

The paper proposes Circuit-based Reasoning Verification (CRV), a method to detect errors in LLM reasoning. The core hypothesis is that reasoning failures might manifest as detectable "structural fingerprints" on a model's computational graph. To study this, the authors modify a transformer model by replacing each MLP with a transcoder (a sparse, interpretable autoencoder trained to imitate the MLP’s function). Using this replacement model, they construct attribution graphs that capture causal connections between tokens, features, and logits during each step of a Chain-of-Thought (CoT).

From these graphs, CRV extracts a set of structural and topological features (e.g., node influence, connectivity, entropy) and trains a diagnostic classifier to predict whether a reasoning step is correct or incorrect. Experiments on synthetic Boolean and arithmetic tasks, as well as GSM8K, suggest that CRV outperforms black-box (e.g., logit-based) and gray-box (e.g., hidden-state probe) baselines. However, the method does not generalise well across reasoning domains, which the authors interpret as evidence that error patterns are domain-specific.

The paper also uses CRV to analyse a single reasoning failure on a simple mathematical expression and show that causal interventions (modifying a single transcoder feature) can correct the models prediction.

**Strengths:**

- The paper's core idea of treating the model's computational graph as a debuggable execution trace is compelling.
- The case study demonstrating the correction of a reasoning error by intervening on a single transcoder feature is interesting.

**Weaknesses:**

- Prior work found that models’ hidden representations often contain detectable traces of reasoning success or failure (e.g., [1, 2]), with simple probes achieving strong performance in similar settings. In contrast, the LR probe baseline reported here performs notably poorly, suggesting it may not have been well-tuned. This undermines confidence in the relative performance gains claimed for CRV.
- CRV combines two major components: (A) replacing dense MLPs with sparse, interpretable transcoders, and (B) constructing and analyzing an attribution graph. The paper attributes its success primarily to (B), yet it never evaluates a simpler baseline that trains a classifier directly on transcoder activation statistics (A) without graph construction. The ablations in Table 3 suggest that “Node Influence & Activation Stats” make up for most of the performance, implying that much of the signal may come from the sparse representation itself.

[1] A. Zhang et al., ‘Reasoning Models Know When They're Right: Probing Hidden States for Self-Verification’, in Second Conference on Language Modeling, 2025.

[2] O. Obeso, A. Arditi, J. Ferrando, J. Freeman, C. Holmes, and N. Nanda, ‘Real-Time Detection of Hallucinated Entities in Long-Form Generation’, arXiv [cs.CL]. 2025.

**Questions:**

NA

---

> ### Author Response · Authors · 2025-11-20
> **Author's rebuttal**
>
> We first would like to thank the reviewers for their feedbacks and comments. Please find below our responses addressing the weaknesses.
>
> _**Weakness 1**_
>
> This is an important point, and our view is that this outcome does not stem from insufficient tuning of the probe. Instead, it highlights a core property of our task that does not appear in the cited settings.
>
> To begin, regarding tuning: our LR Probe results come from an extensive search. As described in Appendix C.2, we trained a separate probe on the hidden states of all 32 layers of the Llama 3.1 8B Instruct model for each dataset, and we chose the best-performing layer for the comparison in Table 1. This procedure gives the probe every advantage available within the limits of a linear method.
>
> Zhang et al. studies a different setup: reasoning models which critique their answers and backtrack if necessary. They segment the reasoning trace such that each segment contains a potential 'final answer'. This is different from our setup where we consider non-reasoning-trained models simply reasoning step-by-step. In our case, each step is an intermediate step towards the final answer, and only the last step contains the final answer. This is a significant difference that means that we should not expect results from Zhang et al. to be reproduced in our setup.
>
> Obeso et al. study a separate problem entirely: predicting whether a specific entity mention is hallucinated. This is a token-level factual check, not a computational or logical correctness task at the level of intermediate steps.
> Finally, to empirically verify that the limitation is not the linearity of the probe, we have added a non-linear MLP Probe baseline in the revised manuscript, and we provide the relevant portion of the result table below (please feel free to check the updated paper for full details). We conducted a grid search over learning rate, weight decay, and hidden dimension size (details added to Appendix C.2).
>
> | Paradigm | Method | AUROC ↑ (Boolean) | AUPR ↑ (Boolean) | FPR@95 ↓ (Boolean) | AUROC ↑ (Arithmetic) | AUPR ↑ (Arithmetic) | FPR@95 ↓ (Arithmetic) | AUROC ↑ (GSM8K) | AUPR ↑ (GSM8K) | FPR@95 ↓ (GSM8K) |
> |----------|---------|-------------------|-------------------|---------------------|------------------------|------------------------|--------------------------|------------------|------------------|--------------------|
> | *Gray-Box* | LR Probe | 52.91 | 0.25 | 88.42 | 54.22 | 1.50 | 91.90 | 55.86 | 7.99 | 90.32 |
> | *Gray-Box* | MLP Probe | 53.63 | 0.26 | 88.56 | 54.41 | 1.30 | 90.98 | 56.02 | 8.63 | 93.94 |
> | **White-Box** | **CRV (Ours)** | **75.87** | **0.97** | **79.17** | **92.47** | **28.92** | **37.09** | **70.17** | **14.3** | **79.61** |
>
> The MLP probe achieved an AUROC of 54.41 on Arithmetic and 56.02 on GSM8K. These results are only marginally better than the linear probe and remain significantly below CRV. This reinforces our hypothesis: the signal for reasoning correctness is not easily accessible in the raw, static hidden states (gray-box), but is imprinted in the causal structure of the computation (white-box). CRV succeeds because it accesses this structural information, which state-based probes miss regardless of their capacity.
>
> _**Weakness 2**_
>
> This is an excellent point that goes to the core of our contribution. We agree that it is important to separate the gains that come from the sparse representation (A) and those that come from the attribution graph (B). Although the feature family is labeled “Node Influence & Activation Stats”, none of these features are computed on the raw transcoder activations. As described in Appendix C.1, every component of this feature family is derived from the attribution graph.
>
> The ablation study therefore does not indicate that the graph can be removed. Instead, it shows that for the tasks we study, local node-level properties extracted from the graph carry more predictive value than global structural properties such as density or connectivity. Both signals depend on the graph, but the local ones appear to be more aligned with the kinds of errors the model makes.

---

> > ### Comment · Reviewer_9q4r · 2025-11-27
> >
> > Thank you for the response and the additional experiments. I remain surprised by the weak performance of the probe but the authors seem to have done sufficient experiments to back these results up. Overall, I like the idea presented in the paper and will raise my score to 8.

---

> > > ### Author Response · Authors · 2025-11-27
> > > **Reply to reviewer's response**
> > >
> > > Thank you for the follow-up and for considering the new experiments. We appreciate your assessment and are glad the added responses addressed your concerns. We are also grateful for the positive view of the idea and for the updated score.

---

### Official Review · Reviewer_Cn2f · 2025-10-31

**Soundness:** 4
**Presentation:** 3
**Contribution:** 4
**Rating:** 8
**Confidence:** 3

**Summary:**

The paper proposes CRV, circuit based reasoning verification, an approach for classifying whether reasoning traces are correct and where they might have possibly made an error. This is done through tracing an attribution graph of an extended retrained formulation of the original LLM. Experiments on boolean, arithmetic and GSM8K reasoning tasks show better performance compared to black- and grey-box methods.

**Strengths:**

Conceptual:
- The idea is very nice and approaches the reasoning veracity problem from a much more fine-grained angle than black- and grey-box  baselines. The overall architecture, i.e. transducers, attribution graph and classifiers on activation and other features are a very nice addition to the mechanistic interpretability community, as far as I know of the literature.

Experimental:
- I like the cross-domain experiment. Even though it shows limited domain generalization, it is a highly useful information and shows the authors adherence to an exhaustive evaluation, even when results are not favourable.
- The intervention is a nice idea. However, it is done on only one single data point (one more in the appendix?) and I wonder about whether you can really identify the reason for incorrect reasoning and do interventions more generally.

Writing:
- The writing is very well done and a pleasure to read.

**Weaknesses:**

Conceptual:
- The method, as acknowledged, is computation heavy and cannot be used without expensive retraining to identify reasoning weaknesses. I think it would be an interesting follow-up question whether such a white-box method could be directly grafted on an existing LLM.
- Some questions on the fairness and experimental setup and comparison to baselines persist, please see below.

**Questions:**

- It is ironic that you need Gradient Boosting, a non-NN learning method, to better understand NNs.
- Is the proposed method also applicable to other LLMs, e.g. Qwen? I do not expect you to perform this experiment, but would like to know your intuition behind that.
- Transcoder:
  - When you replace the original LLM's MLPs by transcoders, how similar are their performance? For example, if the model is significantly altered through that, then whether you can detect faulty reasoning for the surrogate model might not have much to do with detecting it for the original model.
  - Related to the above question: Is the experimental evaluation fair w.r.t. other models? I guess you evaluate other models on the original Llama, while your method is on the modified one. If there is any shift in the MLP, possibly it might become easier overall to detect faulty reasoning.
  - Related to that topic: I have read the appendix on the transcoder training and I am aware that it is done correctly. I still would like to have your opinion on the remaining difference to the original model.
- Could you train a model on all three tasks simultaneously and what would its performance be? From what I understand you train a model for boolean, arithmetic and GSM8K separately.
- Please add an exact formulation for each benchmark metric and how you evaluate all these numbers exactly.

---

> ### Author Response · Authors · 2025-11-20
> **Author's rebuttal (part 1)**
>
> We thank the reviewer for their insightful comments. Please find below our responses to the questions.
>
> _**Q1: It is ironic that you need Gradient Boosting, a non-NN learning method, to better understand NNs.**_
>
> We acknowledge the irony. We believe this validates the strength of our feature extraction rather than signaling a weakness. Our method transforms the opaque, high-dimensional internal state of an LLM into a structured, low-dimensional tabular dataset of graph features. Gradient Boosting is widely considered the state of the art for this type of tabular data. Furthermore, as shown in Appendix C.4, even a simple Logistic Regression model achieves strong performance. This confirms that our white-box approach successfully distills complex reasoning failures into clear, interpretable structural signals that are accessible even to classical machine learning methods.
>
> _**Q2: Is the proposed method also applicable to other LLMs, e.g. Qwen?**_
>
> We expect the CRV framework to apply to Qwen and other Transformer based LLMs because the core architectural elements, specifically the MLP blocks, are standard across these models. While the substitution of these blocks with transcoders is conceptually direct, additional work would be needed to handle specific architectural variants like Mixture of Experts. Furthermore, as our results indicate that error signatures are dependent on the underlying model, the diagnostic classifier would need to be retrained on the new model's data. In summary, the CRV framework offers a general plan for mechanistic analysis of any Transformer model, while still requiring model-specific training for the transcoders and the diagnostic classifier.
>
> _**Q3 & Q4 & Q5 on transcoders**_
>
> The CoT reasoning steps that we verify are generated by the original, unmodified Llama 3.1 Instruct model, not by the surrogate model. Our approach uses the transcoder-augmented surrogate only as an analysis tool to inspect the computation that produced those original steps. The baselines use the hidden states of the original model as their analysis tool.
>
> The evaluation is fair because every method is applied to the same reasoning traces produced by the same source model. The comparison concerns the strength of different verification methods when applied to a fixed output stream. We are not comparing two generative models; we are comparing two analysis procedures applied to the same model.
>
> The transcoder is a high-fidelity approximation of the original MLP but not an exact replica, so some approximation error is unavoidable. This error acts as a form of measurement noise in the analysis tool. We handle this directly: the residual difference is what the “error nodes” in the attribution graphs are meant to represent (see more in [1]). They encode the component of the computation that is not accounted for by the learned sparse features.
>
> [1] https://transformer-circuits.pub/2025/attribution-graphs/methods.html#graphs-constructing

---

> ### Author Response · Authors · 2025-11-20
> **Author's rebuttal (part 2)**
>
> _**Q6:Could you train a model on all three tasks simultaneously and what would its performance be?**_
>
> Thanks for the suggestion. We implemented this experiment and added the results to Table 2 as CRV (Combined), and we have uploaded a revised version of the paper. The performance of this multi-task model generally falls between the zero-shot and in-domain results. While the combined model significantly improves upon the poor zero-shot transfer, it typically lags behind the task-specific "specialist" models. For instance, on the Boolean dataset, the combined model achieves 65.18 AUROC compared to the specialist's 75.87, indicating that learning diverse error signatures simultaneously can introduce interference or capacity constraints. However, in the case of GSM8K, the combined model (70.62) performs on par with the in-domain baseline (70.17), suggesting that for less structured tasks, the added diversity may be beneficial. Overall, this indicates that while a broad-coverage verifier is feasible, there is currently a trade-off between broad generalization and task-specific precision.
>
>
> | Test Set  | Method (Train Set)      | AUROC ↑ | AUPR ↑ | FPR@95 ↓ |
> |-----------|--------------------------|---------|--------|----------|
> | Boolean   | Baseline (MaxProb)       | 58.81   | 0.34   | 95.20    |
> |           | CRV (GSM8K)              | 45.77   | 0.21   | 97.28    |
> |           | CRV (Arithmetic)     | 61.58 | 0.51 | 87.55 |
> |           | CRV (Combined)           | 65.18   | 0.65   | 80.15    |
> |           | *CRV (Boolean)*          | *75.87* | *0.97* | *79.17*  |
> | Arithmetic| Baseline (Energy)        | 76.45 | 5.59 | 73.86 |
> |           | CRV (GSM8K)              | 55.11   | 1.50   | 91.91    |
> |           | CRV (Boolean)            | 69.59   | 2.64   | 72.87 |
> |           | CRV (Combined)           | 90.51   | 30.57  | 44.23    |
> |           | *CRV (Arithmetic)*       | *92.47* | *28.92* | *37.09* |
> | GSM8K     | Baseline (Energy)        | 62.55 | 9.11 | 86.34 |
> |           | CRV (Boolean)            | 44.37   | 6.33   | 95.71    |
> |           | CRV (Arithmetic)         | 57.04   | 7.85   | 94.37    |
> |           | CRV (Combined)           | 70.62   | 12.66  | 80.22    |
> |           | *CRV (GSM8K)*            | *70.17* | *14.3* | *79.61* |
>
> _**Q7: Please add an exact formulation for each benchmark metric and how you evaluate all these numbers exactly.**_
>
> Thank you for this suggestion. We have added the exact mathematical formulations and implementation details for all three metrics (AUROC, FPR@95, AUPR) in Appendix C.3 of the revised paper.

---

### Author Response · Authors · 2025-12-01
**Summary of Revisions and Reviewer Consensus**

Dear AC,

Please find below a summary of the shared points raised during the review process and the specific experiments and revisions we introduced to address them.

**1. Validity of Baselines** (reviewers 9q4r, Mmoa, FZ7t)

Multiple reviewers were initially surprised that our LR probe baseline underperformed compared to black-box methods, questioning if it was sufficiently tuned.
- **Our Response:** We clarified that we performed an exhaustive search across all 32 layers to select the best probe. We explained that, unlike settings such as factual knowledge or hallucination, errors in intermediate steps of reasoning may not leave a linearly separable pattern in static hidden states.
- **New Experiment:** To test whether model capacity was a limiting factor, we implemented and trained a non-linear MLP probe baseline.
- **Result:** The MLP probe performed only marginally better than the LR probe and remained significantly worse than CRV. This supports our view that the signal for reasoning correctness is tied to the causal structure of the computation (captured by CRV) rather than to the distribution of static activations.
- **Paper Update:** We added the MLP Probe details and results to **Appendix C.2** and revised the main results discussion.

**2. Cross-Domain Generalization** (reviewers Cn2f, FZ7t)

Reviewers noted the limits of zero-shot generalization and asked whether a single model could handle multiple domains.
- **New Experiment:** We trained a combined CRV model on the union of all three datasets (Boolean, Arithmetic, GSM8K).
- **Result:** Its performance is generally between the zero-shot and in-domain cases. This suggests that a broad-coverage verifier is possible, but there is a trade-off between broad generalization and task-specific accuracy.
- **Paper Update:** We added the “CRV (Combined)” results to **Table 2** and expanded the analysis in **Section 4.2**.

**3. Metrics and Reproducibility** (reviewer Cn2f)
- **Paper Update:** We added exact mathematical definitions and implementation details for all evaluation metrics (AUROC, FPR@95, AUPR) in **Appendix C.3**.

During the discussion period, reviewers stated that the additional experiments and clarifications addressed their concerns, and two reviewers raised their ratings.

Best regards,

Authors

---

### Meta-Review · Area_Chair_Dbo3 · 2026-01-05

**Summary:**

This work proposes Circuit-based Reasoning Verification (CRV), which utilizes autoencoders to encode MLP layers, constructs computational graphs using the trained autoencoders, detects errors in reasoning paths through these graphs, and corrects them via intervention.

Reviewers highly commend the core concept and contributions of the paper, recognizing the application of mechanistic interpretability techniques to the practical and important task of model debugging and reasoning verification. The paper introduces a novel, fine-grained white-box method that constructs attribution graphs and extracts features to classify the correctness of reasoning steps, while demonstrating its potential for causal discovery through intervention experiments. The experimental design is comprehensive (spanning domains, controlling difficulty levels, and including feature ablation studies), and the writing is clear.

Reviewers have raised the following main concerns:

1. They find it puzzling that some baseline models underperform compared to black-box models.

2. Certain experimental details remain unclear, including how computational graphs are constructed and how errors are diagnosed using these graphs.

**Reviewer Concerns:**

After thorough discussion, the authors have addressed the reviewers' concerns. I believe this work is suitable for an excellent oral presentation.

**Reviewer Scores:**

Reviewer 9q4r has expressed willingness to raise their score from 4 to 8. Reviewer Mmoa has also indicated an intention to increase their score.

---

### Decision · Program_Chairs · 2026-01-26

Accept (Oral)